# Lumina-Next : Making Lumina-T2X Stronger and Faster with Next-DiT

**Le Zhuo**[1,2]* **Ruoyi Du**[1,5]* **Han Xiao**[1,2]* **Yangguang Li**[1,2]* **Dongyang Liu**[1,2]*
**Rongjie Huang**[1]* **Wenze Liu**[1,2]* **Xiangyang Zhu**[1] **Fu-Yun Wang**[2] **Zhanyu Ma**[5]
**Xu Luo**[1] **Zehan Wang**[1] **Kaipeng Zhang**[1] **Lirui Zhao**[1] **Si Liu**[4]
**Xiangyu Yue**[1,2] **Wanli Ouyang**[1,2] **Yu Qiao**[1]† **Hongsheng Li**[2,3]† **Peng Gao**[1]‡*

[1]Shanghai AI Laboratory  [2]The Chinese University of Hong Kong
[3]HKGAI under InnoHK  [4]Beihang University
[5]Beijing University of Posts and Telecommunications

## Abstract

Lumina-T2X is a nascent family of Flow-based Large Diffusion Transformers (Flag-DiT) that establishes a unified framework for transforming noise into various modalities, such as images and videos, conditioned on text instructions. Despite its promising capabilities, Lumina-T2X still encounters challenges including training instability, slow inference, and extrapolation artifacts. In this paper, we present Lumina-Next, an improved version of Lumina-T2X, showcasing stronger generation performance with increased training and inference efficiency. We begin with a comprehensive analysis of the Flag-DiT architecture and identify several suboptimal components, which we address by introducing the Next-DiT architecture with 3D RoPE and sandwich normalizations. To enable better resolution extrapolation, we thoroughly compare different context extrapolation methods applied to text-to-image generation with 3D RoPE, and propose Frequency- and Time-Aware Scaled RoPE tailored for diffusion transformers. Additionally, we introduce a sigmoid time discretization schedule to reduce sampling steps in solving the Flow ODE and the Context Drop method to merge redundant visual tokens for faster network evaluation, effectively boosting the overall sampling speed. Thanks to these improvements, Lumina-Next not only improves the quality and efficiency of basic text-to-image generation but also demonstrates superior resolution extrapolation capabilities and multilingual generation using decoder-based LLMs as the text encoder, all in a zero-shot manner. To further validate Lumina-Next as a versatile generative framework, we instantiate it on diverse tasks including visual recognition, multi-view, audio, music, and point cloud generation, showcasing strong performance across these domains. By releasing all codes and model weights at https://github.com/Alpha-VLLM/Lumina-T2X, we aim to advance the development of next-generation generative AI capable of universal modeling.

## 1 Introduction

Scaling diffusion transformers has unveiled significant improvements in text-conditional image and video generation [15, 14, 31, 35, 62]. Notably, Lumina-T2X [35] introduces flow-based large diffusion transformer (Flag-DiT), which has proven to be a stable, scalable, flexible, and training-efficient architecture for generative modeling across various data domains. For instance, the use

---

*Equal Contribution
†Corresponding Authors
‡Project Lead

38th Conference on Neural Information Processing Systems (NeurIPS 2024).

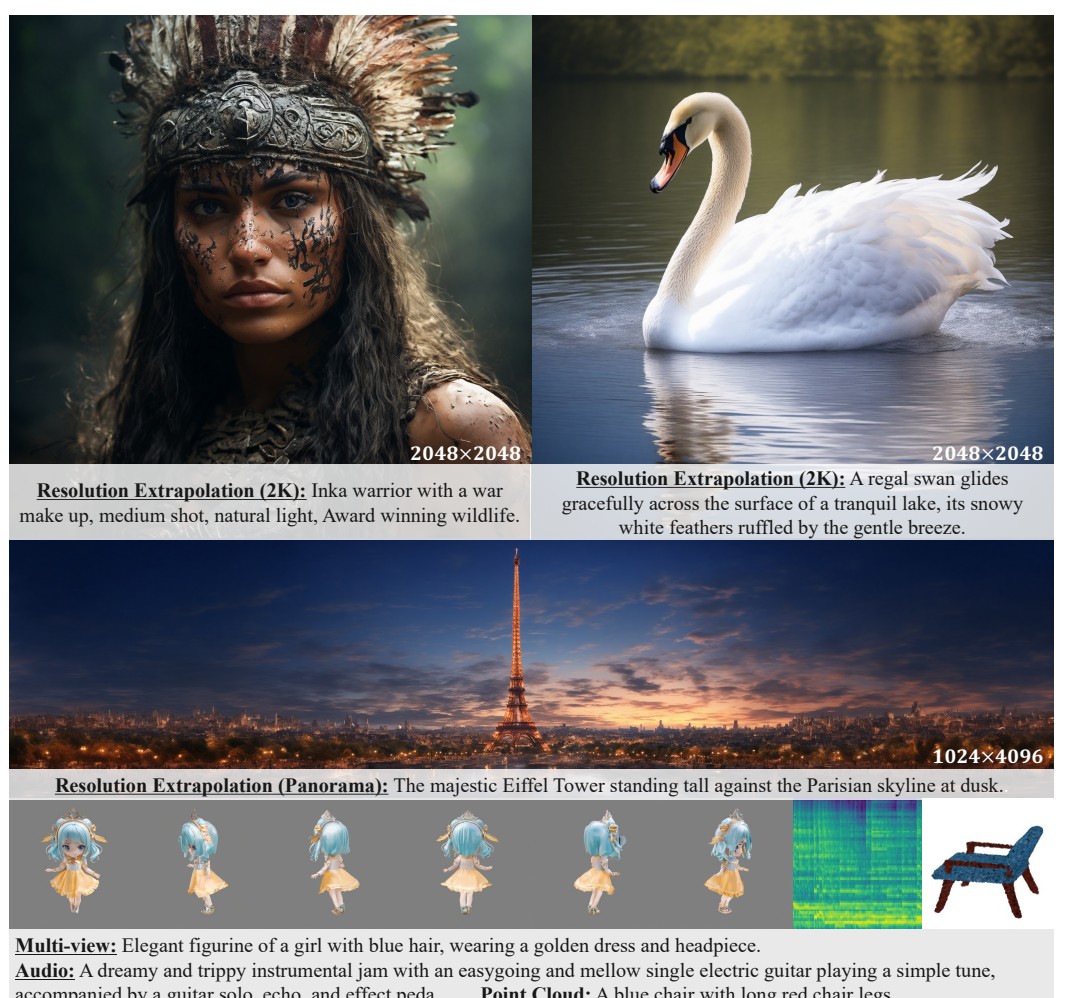

**Resolution Extrapolation (2K):** Inka warrior with a war make up, medium shot, natural light, Award winning wildlife.

**Resolution Extrapolation (2K):** A regal swan glides gracefully across the surface of a tranquil lake, its snowy white feathers ruffled by the gentle breeze.

**Resolution Extrapolation (Panorama):** The majestic Eiffel Tower standing tall against the Parisian skyline at dusk.

**Multi-view:** Elegant figurine of a girl with blue hair, wearing a golden dress and headpiece.
**Audio:** A dreamy and trippy instrumental jam with an easygoing and mellow single electric guitar playing a simple tune, accompanied by a guitar solo, echo, and effect peda.        **Point Cloud:** A blue chair with long red chair legs.

Figure 1: As a foundational generative framework, we demonstrate Lumina-Next's capabilities to generate high-resolution images, multi-view images, general audio and music, and 16K point clouds.

of advanced normalization techniques like KQ-Norm and RMSNorm enable stable mixed-precision training when scaling Lumina-T2X to a 5B Flag-DiT with a 7B LLaMA [80]. Besides, the flow matching framework and Rotary Position Embedding (RoPE) unlock the potential of Lumina-T2X to generate data with arbitrary resolutions, aspect ratios, and durations during sampling. While Lumina-T2X achieves superior visual aesthetics quality with remarkably low training resources, it suffers from weak image-text alignment, slow inference, and extrapolation artifacts due to inadequate training, insufficient training data, and inappropriate context extension strategy.

To fully unleash the potential of scaling diffusion transformers for generative modeling, we present the next generation of Lumina-T2X, Lumina-Next with improved architecture, scaled dataset, optimized sampling techniques, and better context extrapolation strategy, all of which make Lumina-Next stronger and faster. Specifically, the key improvements of Lumina-Next are listed as follows:

**Architecture of Next-DiT** We revisit the architecture design of Flag-DiT and find suboptimal components for scalable generative modeling in the visual domain. First, we replace the 1D RoPE with 3D RoPE to eliminate the inappropriate positional priors to model images and videos using the attention mechanism. We further remove all learnable identifiers in Flag-DiT, such as [nextline] and [nextframe], since our 3D RoPE already provides sufficient 3D positional information. Next, we task an in-depth analysis of the instability during both training and sampling and find it stems from the uncontrollable growth of network activations across layers. Therefore, we introduce the sandwich normalization block [25] in attention modules, which is proven to effectively control the activation magnitudes. Additionally, we employ the Grouped-Query Attention [4] to reduce com-

putational demand especially when generating high-resolution images. The improved architecture of Next-DiT is validated on the ImageNet-256 benchmark, demonstrating a faster convergence rate compared to both Flag-DiT and SiT [59].

**Frequency- and Time-Aware Scaled RoPE** Inspired by the recent progress of context extrapolation in LLMs [18, 1, 64], Lumina-T2X directly applies NTK-Aware Scaled RoPE to generate higher resolution images thanks to the design of 1D RoPE. However, a comprehensive analysis of using 3D RoPE for length extrapolation in the vision domain is still lacking, especially tailored for diffusion and flow models. To bridge this gap, we first conduct a holistic comparison between existing extrapolation methods applied to text-to-image diffusion transformers, including Position Extrapolation, Position Interpolation [18], and NTK-Aware Scaled RoPE [1]. We rethink the relationship between the encoding frequency in RoPE and the generated content and propose a Frequency-Aware Scaled RoPE, significantly reducing content repetition during extrapolation. Furthermore, considering the difference between the auto-regressive generation of LLMs and the time-aware generation process with fixed sequence length in diffusion models, we propose a novel Time-Aware Scaled RoPE for diffusion transformers to generate high-resolution images with global consistency and local details.

**Optimized Time Schedule with Higher-Order Solvers** In contrast to various time schedules [61, 74, 42] and advanced SDE/ODE solvers [53, 54, 26, 42, 81] in diffusion models, most flow models [45, 50, 31] still adopts Euler's method with uniform time discretization during sampling. SiT [59] conducts a preliminary exploration of using different diffusion schedules and higher-order ODE solvers on the ImageNet benchmark. In this work, we demonstrate that existing diffusion schedules are not suitable for flow models and propose novel time schedules tailored for flow models to minimize discretization errors. We further combine the optimized schedules with higher-order ODE solvers, achieving high-quality text-to-image generation samples within only 5 to 10 steps.

**Time-Aware Context Drop** To reduce the network evaluation time of each single step, we propose dynamically merging latent tokens to reduce redundancy in attention blocks. Different from the complex merge-and-unmerge algorithm in Token Merging [9], we employ a simple average pooling for keys and values to aggregate similar context tokens in spatial space. Additionally, we augment this Context Drop with time awareness to align with the dynamic sampling process of diffusion models. The resulting Time-Aware Context Drop is validated to effectively enhance inference speed while maintaining visual quality, achieving a $2\times$ inference-time speed up in 1K resolution image generation. When integrated with advanced attention inference techniques like Flash Attention [22], our method further improves inference speed of generating ultra-high resolution images.

By composing all improvements over Lumina-T2X, we reach Lumina-Next. Our Lumina-Next, featuring a 2B Next-DiT and Gemma-2B as text encoder, achieves better text-to-image generation compared to Lumina-T2X with a 5B Flag-DiT and LLaMA-7B [80] as text encoder, while significantly reducing training and inference costs. Unlike previous models [67, 65, 31, 15, 14] that rely on CLIP [66] or T5 [20], Lumina-Next demonstrates strong zero-shot multilingual ability with LLMs as text encoder. Moreover, equipped with improved architecture and the proposed inference techniques, Lumina-Next can achieve tuning-free 2K and few-step generation. We also show that Lumina-Next's framework can be easily extended to other modalities with excellent performance, demonstrating that Lumina-Next is a unified, versatile, powerful, and efficient framework for generative modeling. All codes and checkpoints of Lumina-Next are released. We hope the reproducibility of Lumina-Next can foster transparency and innovations in the generative AI community.

## 2 Improving Lumina-T2X

### 2.1 Architecture of Next-DiT

In Lumina-T2X [35], Flag-DiT serve as the core architecture with flow matching formulation, ensuring training stability and scalability for larger models. In this section, we propose Next-DiT, an improved version of Flag-DiT with the following key modifications. The comparison of detailed architectures of Next-DiT and Flag-DiT can be found in the Appendix B.1.

**Replacing 1D RoPE and Identifiers with 3D RoPE** Flag-DiT replaces absolute positional embedding (APE) with RoPE to enable flexibility of extrapolating to unseen resolution during training.

It further introduces learnable tokens including [nextline] and [nextframe] as identifiers to model images and videos with different aspect ratios and durations. We discovered that 1D RoPE is a lossy representation to encode accurate spatial-temporal positions and the learnable identifiers introduce additional design complexity, all of which can be simplified to 3D RoPE without losing any information.

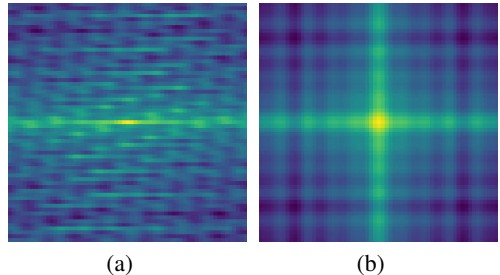
(a)          (b)

Figure 2: Visualization of attention score using (a) 1D RoPE and (b) 2D RoPE on images. We set the central point in the image as the anchor query.

Specifally, Lumina-T2X encode input from different modalities into latent frames of shape $[H, W, T, C]$, where $T = 1$ for images, $T =$ numframes for videos, and $T =$ numviews for multi-view images. Then, 1D RoPE is applied to the flattened 1D token sequences, which is formulated as follows given the $m$-th query and $n$-th key $q_m, k_n \in \mathbb{R}^{d_{\text{head}}}$:

$$\tilde{q}_m = f(q_m, m) = q_m e^{im\Theta}, \quad \tilde{k}_n = f(k_n, n) = k_n e^{in\Theta}, \tag{1}$$

where $\Theta = \text{Diag}(\theta_1, ..., \theta_{d_{\text{head}/2}})$ is the frequency matrix. However, 1D RoPE directly treats flattened frame tokens as 1D language sequences, discarding the spatial and temporal relations between different positions. As shown in Figure 2(a), when applying 1D RoPE to the 2D images, the resulting long-term decaying property is obviously incorrect for the 2D scenario. Inspired by recent works that introduce 2D RoPE in the image domain [55, 32], it is natural to extend the original 1D RoPE to 3D RoPE for spatial-temporal modeling. We divide the embedding dimensions into three independent parts and calculate positional embedding for the $x$-axis, $y$-axis, and $z$-axis separately. Given then 3D coordinates of the $m$-th query and $n$-th key, the 3D RoPE is defined as:

$$\begin{aligned} \tilde{q}_m &= f(q_m, t_m, h_m, w_m) = q_m[e^{it_m\Theta_t} \parallel e^{ih_m\Theta_h} \parallel e^{iw_m\Theta_w}], \\ \tilde{k}_n &= f(k_n, t_n, h_n, w_n) = k_n[e^{it_n\Theta_t} \parallel e^{ih_n\Theta_h} \parallel e^{iw_n\Theta_w}], \end{aligned} \tag{2}$$

where $\Theta_t = \Theta_h = \Theta_w = \text{Diag}(\theta_1, ..., \theta_{d_{\text{head}/6}})$ is the divided frequency matrix for each axis and $\parallel$ denotes concatenating complex vectors of different axes at the last dimension. The attention scores with 3D RoPE are calculated by taking the real part of the standard Hermitian inner product:

$$\begin{aligned} &\text{Re}[f(q_m, t_m, h_m, w_m)f^*(k_n, t_n, h_n, w_n)] \\ &= \text{Re}[q_m k_n^* e^{i\Theta_t(t_m - t_n)} \parallel q_m k_n^* e^{i\Theta_h(h_m - h_n)} \parallel q_m k_n^* e^{i\Theta_w(w_m - w_n)}], \end{aligned} \tag{3}$$

where $\text{Re}[\cdot]$ denotes the real part of complex numbers and $^*$ indicates complex conjugates.

With the introduction of 3D RoPE, we provide a unified and accurate spatial-temporal representation of positional encoding for different modalities, without the need to add extra learnable tokens such as [nextline] and [nextframe] to indicate the row, column, and frame index. Our 3D RoPE also incorporates the 2D RoPE formulation since it naturally becomes 2D RoPE when applied to images with only 1 frame. Figure 2 shows the 2D RoPE assigns high attention scores to positions within the same row and column, which is a natural prior for images. Furthermore, our decoupled implementation of 3D RoPE on spatial and temporal axes better unlocks the potential for length extrapolation when combined with inference-time techniques mentioned in Section 2.2. We demonstrate the superior zero-shot extrapolation results with arbitrary resolutions in Section 3.1.

**Controling Magnitiude of Network Activations** To stabilize the training process when scaling up model size and token length, Lumin-T2X replaces all LayerNorm with RMSNorm and add QK-Norm before key-query dot product attention computation. Though effective to some extent, this recipe is insufficient for stabilizing the training of extremely large diffusion transformers on long sequences. A closer examination of the Flag-DiT architecture reveals the presence of long signal paths without any normalization due to the residue structure of attention and mlp layers. We argue that these unnormalized paths appeared in many diffusion transformer architectures [63, 15, 59] can accumulate signal values across layers, causing uncontrollable growth of network activations, especially in deeper layers. We validate this hypothesis by visualizing the evolution of activation magnitudes over different depths of the text-to-image model using 500 random samples at different timesteps, as shown in Figure 3. What's worse, the uncontrollable network activations not only lead

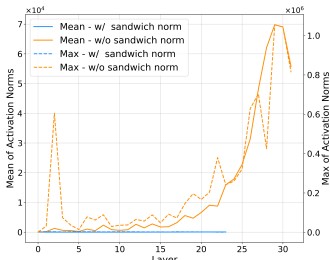
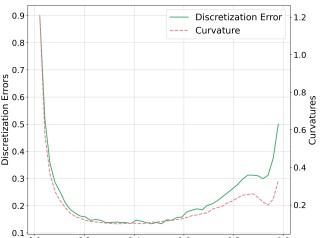
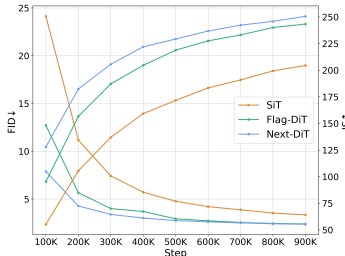

Figure 3: Sandwich normalization effectively controls activation magnitudes over layers.

Figure 4: Discretization errors and local curvatures grow at the start and end of sampling.

Figure 5: Next-DiT converges faster than SiT and Flag-DiT in terms of FID and IS.

to an unstable training process but also deteriorate the sampling process. Particularly for resolution extrapolation, small errors in the early layers or sampling steps are amplified in subsequent layers or steps. These compounding errors gradually cause extremely out-of-distribution inputs for the network, leading to failure in generated samples.

Therefore, we propose a simple yet effective approach that adds RMSNorm both before and after each attention and mlp layer. The modified architecture is equivalent to using the *sandwich normalization* [25] in transformer blocks. Importantly, both pre-norm and post-norm are placed before the scale operation in AdaLN-Zero to prevent normalizing an all-zero tensor at the start of training, which is caused by the zero-initialized scale parameter in AdaLN-Zero. However, after the scale operation, the activation magnitudes still cannot be preserved. We further add a tanh gating to the second scale prediction after post-norm to prevent extremely large modulation values from contributing to the residual branch. With all these efforts including the original QK-Norm in Flag-DiT, we effectively control both then input and output activation magnitudes of each attention and mlp layer, as validated in Figure 3.

**Experiments on ImageNet** To quantitatively assess the effects of Next-DiT with the above improvements, we conduct experiments on the label-conditional ImageNet-256 benchmark. We follow the training setups and evaluation protocols of SiT [59] and Flag-DiT [35]. As depicted in Figure 5, Next-DiT converges significantly faster than both Flag-DiT and SiT evaluated by FID and Inception Score (IS). This observation confirms that the improved design of Next-DiT can enhance its fundamental generative capabilities. Additionally, we ablate the effectiveness of long-skip connections[6] designed to provide shortcuts for low-level features. Unfortunately, long-skip connections lead to training instability and significantly worsen the performance of Next-DiT. Therefore, Next-DiT chooses to not incorporate long-skip connection.

## 2.2 Improving NTK-Aware Scaled RoPE with Frequency- and Time-Awareness

In the large language models (LLMs) community, tuning-free or few-shot tuning length extrapolation is a popular focus currently and has been thoroughly analyzed, *e.g.*, position interpolation [18], NTK-Aware Scaled RoPE [1], YaRN [64], *etc*. In Lumina-T2X, due to the adoption of 1D RoPE same as LLMs, NTK-Aware Scaled RoPE has been directly applied to achieve a certain degree of resolution extrapolation. In Lumina-Next, we further investigate the effects of different length extrapolation methods on 3D RoPE for the first time. Based on our findings, we introduce frequency- and time-awareness grounded in NTK-Aware Scaled RoPE, resulting in novel Frequency-Aware Scaled RoPE and Time-Aware Scaled RoPE targeted to the visual generation task. In Figure 6 (a), we demonstrate the wavelength of each dimension of the RoPE embedding under different extrapolation techniques with a toy setting of $b = 5$, $h_{\text{head}} = 24$, and $T = H = W = 16$.

**Revisiting NTK-Aware Scaled RoPE** 3D RoPE encodes position information of each axis using a frequency matrix $\Theta = \text{Diag}(\theta_1, \cdots, \theta_d, \cdots, \theta_{d_{\text{head}}/6})$ with $\theta_d = b^{-6d/d_{\text{head}}}$, where $b$ is the rotary base. When we want to perform a $s$ times resolution extrapolation, the most straightforward way is encoding unseen positions with no change of RoPE, known as **Position Extrapolation**. However, these unseen positions will directly confuse the model's spatial understanding, leading to the generation of unreasonable or repetitive content (refer to Figure 6 (c)).

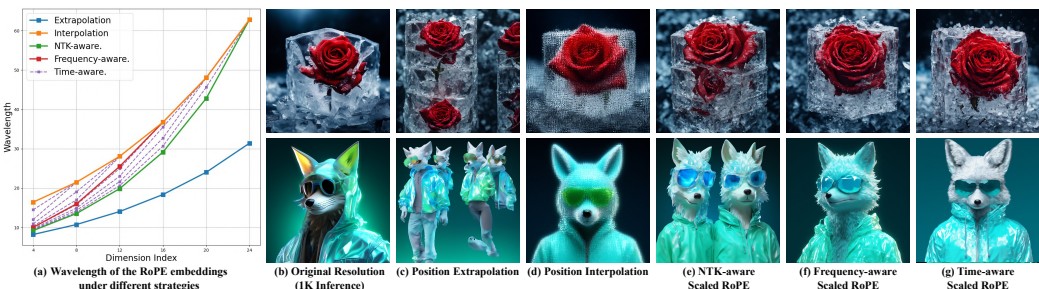

(a) Wavelength of the RoPE embeddings under different strategies    (b) Original Resolution (1K Inference)    (c) Position Extrapolation    (d) Position Interpolation    (e) NTK-aware Scaled RoPE    (f) Frequency-aware Scaled RoPE    (g) Time-aware Scaled RoPE

Figure 6: (a) Toy illustration of RoPE embeddings' wavelength with different extrapolation strategies. (b)-(g) Results of different resolution extrapolation strategies of 2K generation.

In contrast, another straightforward approach, **Position Interpolation** [18], involves linearly rescaling the frequency matrix as $\theta_d^{\text{inter}} = \theta_d \cdot s$, thereby bringing all position embeddings back into the training value domain. However, as noted by [1], this linear interpolation makes it difficult for RoPE to distinguish between positions of tokens that are very close to each other. The same conclusion exists in the diffusion transformer – as shown in Figure 6 (d), the global structure of the image is maintained at the cost of losing all local details.

To address these issues, NTK-Aware Scaled RoPE [1] is proposed to achieve training-free length extrapolation for LLMs. Particularly, it scales the rotary base as $b' = b \cdot s$ and then calculates $\theta_d^{\text{ntk}} = b'^{-6d/d_{\text{head}}}$ such that the lowest frequency term of RoPE is equivalent to performing position interpolation, allowing a gradual transition from position extrapolation of high-frequency terms to position interpolation of low-frequency ones. In Lumina-T2X [35], NTK-Aware Scaled RoPE has been shown to seamlessly integrate with diffusion transformers to achieve resolution extrapolation to a certain extent. However, failure cases still occur, such as the repetition issue shown in Figure 6 (e), indicating that the model still struggles with spatial awareness during resolution extrapolation.

**Frequency-Aware Scaled RoPE**    Rethinking the limitations of NTK-Aware Scaled RoPE, we believe that performing interpolation on the lowest frequency component is a suboptimal design. Considering that RoPE uses periodic functions with frequencies $\Theta$ to encode relative positions, many dimensions encounter unseen repeated cycles during resolution extrapolation, which in turn causes the model to output repetitive content. This echoes the statement of YaRN [64] that position embedding should be divided into parts using wavelength as the boundary and treated differently. To solve this problem, we first identify the dimension with the wavelength equivalent to the training sequence length as $d_{\text{target}} = d_{\text{head}} \cdot \log_b(\frac{L}{2\pi})$, where $L$ is the maximum training sequence length. After that, a scaled base with frequency-awareness can be formulated by $b' = b \cdot s^{\frac{d_{\text{head}}}{d_{\text{target}}}}$, which enable the component at $d_{\text{target}}$ equivalent to position interpolation. Note that $d_{\text{target}}$ may not be an integer, but we do not need to use it as an index so that we can use its exact value. Additionally, to prevent the part where $d > d_{\text{target}}$ from being excessively interpolated, we take the larger value comparing with the position interpolation result as $\theta_d^{\text{freq}} = \max(b'^{-6d/d_{\text{head}}}, b^{-6d/d_{\text{head}}} \cdot s)$. This Frequency-Aware Scaled RoPE further addresses the issue of content repetition, as shown in Figure 6 (f).

**Time-aware Scaled RoPE**    Observing Figure 6, we can conclude that: (1) while position interpolation confuses local details, it effectively ensures a reasonable global structure, and (2) while NTK-Aware Scaled RoPE has content repetition, it retains good local details. By combining these observations with the time-conditioned feature of diffusion models, we can achieve a more targeted resolution extrapolation approach. A common understanding of diffusion models is that they first restore global concepts before local details [19]. This motivates us to design a time-aware strategy to benefit from different approaches at different denoising steps – using position interpolation in the early stages of denoising to ensure the overall structure and gradually shifting to NTK-Aware Scaled RoPE to preserve local details. This echoes DemoFusion's [29] strategy of gradually reducing global guidance over the denoising process, allowing the model to focus on adding local details. Specifically, the previously proposed Frequency-Aware Scaled RoPE is naturally adaptable to this progressive shift. When $d_{\text{target}} = 1$, it is equivalent to position interpolation; when $d_{\text{target}} = d_{\text{head}}$, it is equivalent to NTK-Aware Scaled RoPE. Therefore, we design a coefficient $d_t$

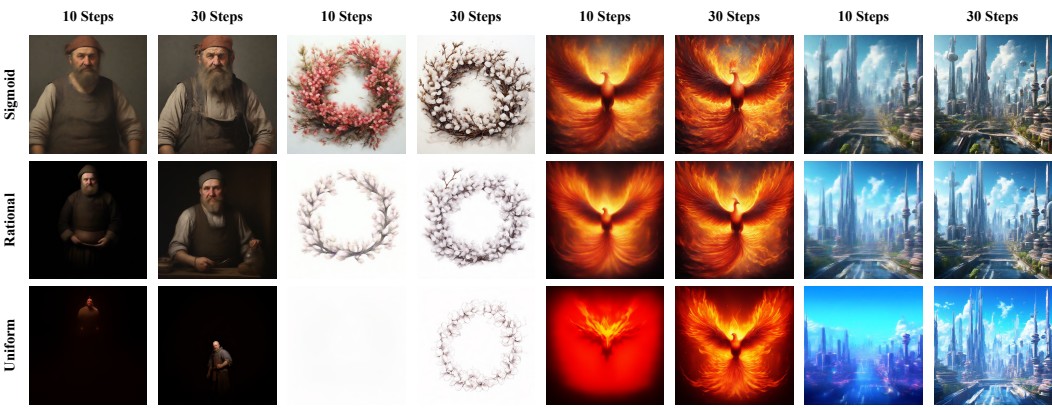

Figure 7: Comparison of few-step generation using different time schedules with Euler's method.

that varies with time $t$ in the most direct way, making the frequency base time-conditioned, *i.e.*, $b'_t = b \cdot s^{\frac{d_{\text{head}}}{d_t}}$, where $d_t = (d_{\text{head}} - 1) \cdot t + 1$. Then the time-aware frequency base can be expressed as $\theta_{d,t}^{\text{time}} = max(b'^{-6d/d_{\text{head}}}_t, b^{-6d/d_{\text{head}}} \cdot s)$. This Time-Aware Scaled RoPE makes globally consistent high-detail resolution extrapolation possible, as shown in Figure 6 (g). In the following experiments on resolution extrapolation, we use Time-Aware Scaled RoPE for the best tuning-free performance.

## 2.3 Improving Sampling Efficiency with Fewer and Faster Steps

Different from one-step generative models, the sampling of diffusion and flow models involves an iterative denoising process. The overall time complexity of sampling can be written as $N \times T$, which is controlled by the number of function evaluations $N$ and the inference time of a single function evaluation $T$. Therefore, to improve the efficiency during inference, we propose corresponding strategies for each factor to reduce $N$ and $T$ respectively.

**Fewer Sampling Steps with Optimized Time Schedule and Higher-Order Solvers**    Both diffusion models and flow-based models numerically solve an SDE or ODE within the interval $[t_{min}, t_{max}]$ during sampling. A common approach is to discretize the continuous time interval into multiple sub-intervals and use Euler's method with $\mathcal{O}(h^2)$ discretization error with respect to step size $h$ or other advanced solvers. Given a constant budget of $N$ sampling steps, the time discretization strategy $\{t_0, t_1, ..., t_N\}$, or time schedule, determines the grid of timesteps on which the network is evaluated. The resulting accumulated truncation errors can greatly impact the quality of generated samples, especially when $N$ is small. Various time schedules have been explored for diffusion models, such as Cosine [61], Linear [74], and EDM [42]. However, most flow-based models still adopt the simplest uniform time intervals and first-order Euler's method for sampling, resulting in poor-quality images in low-resource scenarios, as shown in Figure 7.

Therefore, we provide a detailed analysis of truncation errors at different timesteps using Euler's method to solve the Flow ODE for text-to-image generation. We begin by setting the anchor sampling steps, $N = 50$, with uniform intervals for evaluation. For each timestep $t_i$, the sample produced by one-step Euler's method is defined as $\hat{\mathbf{x}}_{t_{i+1}} = \mathbf{x}_{t_i} + (t_{i+1} - t_i)\mathbf{v}_\theta(\mathbf{x}_{t_i}, t_i)$. To approximate the local truncation error $\tau_{t_i}$ at time $t_i$, we estimate the ground truth sample $\mathbf{x}_{t_{i+1}}$ by running a large number of Euler steps from $t_i$ to $t_{i+1}$ with uniform intervals and compute the discretization error as $\tau_{t_i} = \mathbf{x}_{t_{i+1}} - \hat{\mathbf{x}}_{t_{i+1}}$. In addition, since the flow model builds linear interpolation between noise and data and predicts the constant velocity field, it is natural to use the curvature to describe local errors on the trajectory, *i.e.*, the straighter the trajectory, the smaller the discretization error. Following [60], we compute the second-order derivative to estimate the local curvature of each step as $\kappa_{t_i} = \mathbf{x}_{t_i} - (\mathbf{x}_{t_{i+1}} + \mathbf{x}_{t_{i-1}}) \times 0.5$. The average L2-norm of discretization error $\tau_{t_i}$ and curvature $\kappa_{t_i}$ are visualized in Figure 4. Our analysis reveals that discretization errors are much larger when $t$ is near pure noise ($t = 0$) and are relatively larger near clean data ($t = 1$), compared to intermediate timesteps. This observation contradicts the findings in diffusion models, where EDM [42] discover

the discretization errors monotonically increase as samples get closer to clean data. Hence, diffusion schedules like EDM are not ideal for flow-based models.

To optimize the time schedules for sampling, we propose two parametric functions to map $t \in [0, 1]$:

$$\text{RATIONAL} \quad t' = \frac{t}{\sigma - t + \sigma t}, \qquad \text{SIGMOID} \quad t' = \begin{cases} \frac{1}{1 + \exp(-\alpha(t-\mu))} & \text{if } t < \mu \\ 1 - \frac{1}{1 + \exp(\beta(t-\mu))} & \text{if } t \geq \mu \end{cases}. \quad (4)$$

These time schedules are tailored for flow-based models. The first, a rational function parametrized by $\sigma$, maps all timesteps to smaller values, minimizing the discretization errors near pure noise. This schedule reduces to the uniform time schedule when $\sigma = 1$ and is also used in prior works [35, 31, 38] for high-resolution image generation. The second schedule, a piecewise sigmoid-like function, ensures that the step sizes at the start and end of sampling are larger than those at intermediate steps. The central point is controlled by $\mu$, while the weights on lower and higher timesteps are adjusted by $\alpha$ and $\beta$, respectively. As shown in Figure 7, both rational and sigmoid schedules improve few-step sampling performance compared to all-black or all-white images generated by the uniform schedule. This highlights the importance of the early stage of sampling in creating the main subject in the image. The sigmoid schedule generates images with better details and is adopted for the remainder of this paper with $\mu = 0.6$, $\alpha = 6$, and $\beta = 20$.

Notably, this optimized schedule requires no extra computation but a free launch for fast sampling using our flow model. It is also compatible with advanced solvers applicable to flow models, which can unleash its full potential. However, limited work has explored the use of higher-order solvers for solving the Flow ODE. Unlike the Probability Flow ODE [75] in diffusion models, which has a semi-linear structure, the Flow ODE derived from the continuity equation is in a simpler form of $\dot{\mathbf{x}} = \mathbf{v}_\theta(\mathbf{x}_t, t)$. This simplicity allows for the direct use of existing higher-order ODE solvers, such as the traditional explicit Runge-Kutta (RK) methods in the form of $\mathbf{x}_t = \mathbf{x}_s + \int_s^t \mathbf{v}_\theta(\mathbf{x}_\tau, \tau)\mathrm{d}\tau$, where the $n$-th order solver ensembles $n$ intermediate steps between $[s, t]$ to reduce the approximation errors. During sampling, explicit RK family solvers also need to specify the timesteps $\{t_0, t_1, ..., t_N\}$ in advance and adopt the uniform intervals as default. Thus, our improved schedules, such as the rational or sigmoid schedule, can be directly applied to these higher-order solvers. Algorithm B.1 illustrates the pseudocode for combining the midpoint method and sigmoid schedule for sampling.

**Faster Network Evaluation with Time-Aware Context Drop**    After reducing the denoising steps during sampling, our next goal is to optimize the network evaluation time $T$ for each single step, which is constrained by the quadratic complexity of attention blocks. Among various inference-time acceleration methods, Token Merging (ToMe) [9] is a simple yet effective approach that merges similarly visual tokens to reduce redundancy in attention blocks, thereby speeding up inference for image classification. Although ToMe has been successfully applied to diffusion U-Net [10, 73], we find that a straightforward application of ToMe to Next-DiT, a purely transformer-based diffusion model, completely fails to produce satisfactory images. We hypothesize that performing the original token merging in all transformer layers across all diffusion steps is clearly unsuitable for the task of generative modeling with diffusion transformers.

To bridge this gap, we propose Time-Aware Context Drop, an optimized method to drop redundant visual tokens during inference for Next-DiT. First, the original bipartite soft matching in ToMe not only introduces additional complexity but also contradicts the design of 3D RoPE in Next-DiT. Therefore, we replace this complex token partitioning strategy in ToMe with a simple average pooling for downsampling, aligning with the spatial prior introduced in 3D RoPE. Similar to [73], we find that applying the merge-and-unmerge operation to all queries, keys, and values greatly undermines the quality of generated images as this strategy is designed for visual recognition. To preserve the complete visual content, we only perform average pooling on keys and values for downsampling. Finally, considering the time-awareness in diffusion models, we perform token dropping for all layers at $t = 0$ to enhance efficiency and gradually transform to no token dropping at $t = 1$ to maintain visual quality. The Context Drop without time-awareness is similar to the key-value (KV) token compression in PixArt-$\Sigma$ [14], which merges KV tokens in spatial space using $2 \times 2$ convolution to reduce complexity. However, unlike their fine-tuning methods, Context Drop is a training-free technique with an adaptive compression rate that aligns with the diffusion time schedule. In Appendix C.1, We demonstrate samples generated by our methods, which reduce inference time while achieving comparable visual quality.

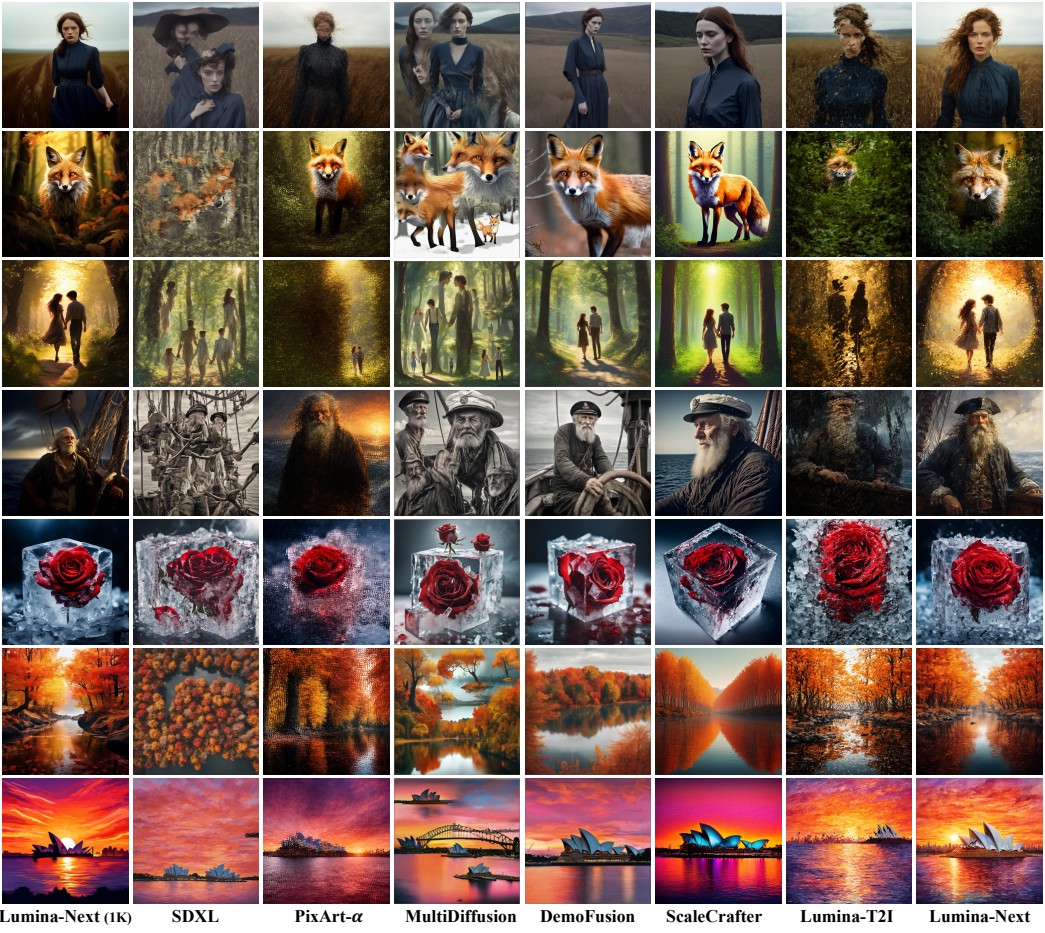

| Lumina-Next (1K) | SDXL | PixArt-α | MultiDiffusion | DemoFusion | ScaleCrafter | Lumina-T2I | Lumina-Next |

Figure 8: Comparison of $4\times$ resolution extrapolation.

# 3 Lumina-Next by Composing Everything

By composing all improvements, we finally reached Lumina-Next, a powerful generative framework for various modalities with improved efficiency. We first instantiate Lumina-Next for text-to-image generation, which is stronger and faster than Lumina-T2X. Besides, compared with SDXL and PixArt-$\alpha$, it demonstrates preliminary supports of multilingual prompts (refer to Appendix C.2), better visual quality with smaller sampling steps, and more flexibility at synthesizing images with any resolution. After this, we further verified that the Lumina-Next framework also exhibits excellent performance in any-resolution recognition (refer to Appendix D) and multi-view images, music, audio, point clouds generation (refer to Appendix E).

## 3.1 Flexible Resolution Extrapolation

**Failure of SDXL and PixArt-$\alpha$**  SDXL [65] and PixArt-$\alpha$ [15] are representative methods of U-Net and transformer-based diffusion models, respectively. Despite their ability to generate impressive images, they fail to produce images beyond the training resolution, resulting in repetitive and unreasonable content at higher resolutions, as shown in Figure 8. This is because the receptive field of U-Net and the absolute position embedding of transformers are significantly biased towards the training resolution or sequence length.

**Comparison with Tuning-free Resolution Extrapolation Methods**  Due to the enormous cost of directly training high-resolution (greater than 2K) generative models, researchers have sought to enable pre-trained diffusion models to infer beyond the training resolution in a tuning-free manner. MultiDiffusion [7] is the first attempt to generate high-resolution panoramas via sampling overlapped sliding windows at the original resolution. DemoFusion [29] further utilizes the generated

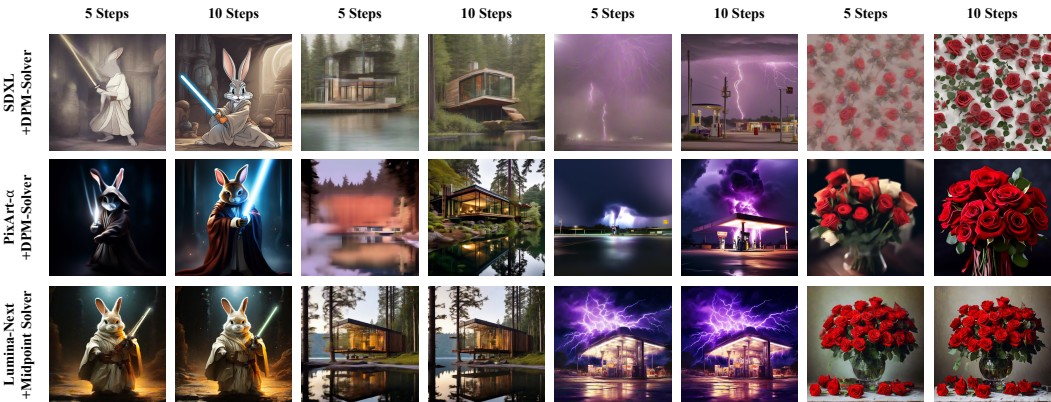

Figure 9: Comparison of few-step generation using second-order ODE solvers.

results at training resolution as global guidance, achieving object-oriented high-resolution image generation. Meanwhile, ScaleCrafter [37] adopts dilated convolution to achieve resolution extrapolation tiled for U-Net based diffusion models (*e.g.*, SDXL [65]).

In Figure 8, we compare Lumina-Next with the methods above on the 2K generation task. We can conclude that MultiDiffusion, proposed for panorama generation, lacks global awareness, leading to the issue of repetitive content still being present. DemoFusion and ScaleCrafter work well, but DemoFusion requires multiple runs, resulting in a non-negligible inference budget, and ScaleCrafter is built upon dilated convolution, which is unsuitable for modern transformer-based diffusion models. In contrast, Lumina-Next inherently supports a direct resolution extrapolation with the carefully designed Next-DiT architecture and frequency- and time-aware scaled RoPE at inference time, offering a more fundamental solution in the era of diffusion transformer.

**On the Importance of Next-DiT for Resolution Extrapolation** Regarding Figure 8, we particularly emphasize that Lumina-Next's 2K extrapolation performance significantly surpasses that of Lumina-T2X. In addition to the newly proposed time-aware scaled RoPE, we attribute this to the advanced design of Next-DiT: (1) 3D RoPE enables independent modelling of the $t$, $h$, $w$ dimensions, decoupling extrapolation in three dimensions so that they do not affect each other, and (2) sandwich norm stabilizes the output magnitude of each transformer block, reducing the model's sensitivity to sequence length. Composing them together, Lumina-Next possesses excellent resolution extrapolation capabilities without the need for any special inference algorithms like DemoFusion [29].

## 3.2 Few-Step Text-to-Image Generation

We demonstrate how our improved sampling schedules together with higher-order ODE solvers can significantly boost the sampling efficiency for flow models. Specifically, we employ the midpoint solver, a second-order Runge-Kutta method, combined with our sigmoid schedule following Algorithm B.1. For comparison, we choose open-source text-to-image models, including SDXL [65] and PixArt-$\alpha$ [15] equipped with DPM-Solver [53, 54]. Figure 9 shows Lumina-Next using the midpoint solver significantly improves the sample quality in 10-20 number of function evaluations (NFEs), consistently achieving better conditional sampling performance compared to PixArt-$\alpha$and SDXL using DPM-Solver. Switching to higher-order solvers can further enhance few-step sampling performance with the cost of linearly increasing NFEs, presenting a speed-quality trade-off.

## 4 Conclusion

In this paper, we introduce Lumina-Next, which successfully addressed the limitations of its predecessor, Lumina-T2X. With the improved Next-DiT architecture, context extrapolation together, and fast sampling techniques tailored for our flow-based diffusion transformers, Lumina-Next showcases strong generation capabilities, such as generating high-quality images significantly larger than its training resolution and multilingual text-to-image generation. Remarkably, these results are all produced in a training-free manner and outperform previous methods including SDXL and PixArt-$\alpha$. We extend Lumina-Next's versatility to other modalities, such as multiviews, audio, music, and point clouds, with minimal modifications, achieving superior results across these diverse applications.

# 5  Acknowledgements

This project is funded in part by the National Key R&D Program of China Project 2022ZD0161100, by the Centre for Perceptual and Interactive Intelligence (CPII) Ltd under the Innovation and Technology Commission (ITC)s InnoHK, by the General Research Fund of Hong Kong RGC Project 14204021, by the National Natural Science Foundation of China (Grant No. 62306261), and by the Shun Hing Institute of Advanced Engineering (SHIAE) No. 8115074. Hongsheng Li is a PI of HKGAI under the InnoHK.

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

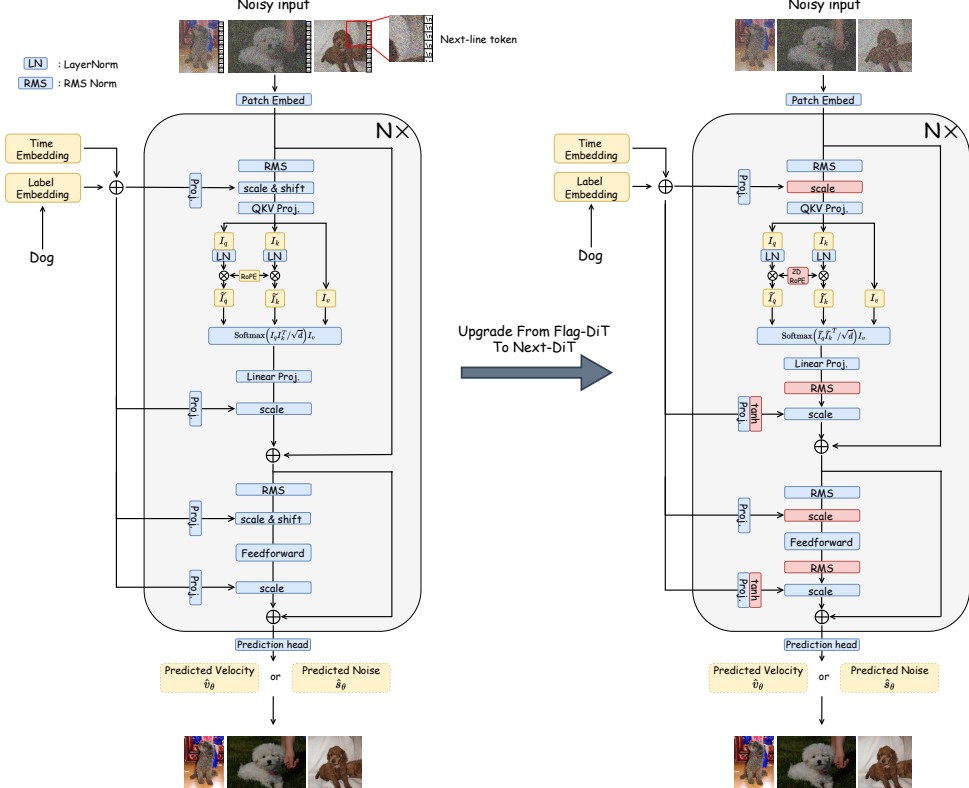

Figure 10: Architecture details of Flag-DiT and Next-DiT.

# A  Broader Impacts and Limitations

## A.1  Broader Impacts

In this work, we propose a fundamental framework for generative modeling. On the positive side, the open-source of our code and checkpoints will foster transparency as well as innovations in the generative AI field. On the negative side, our text-to-image generation model, for example, may be used to generate images with potential biases and malicious information, similar to any generative AI tool. However, we have designed and used a filtering pipeline to filter out all malicious content in the training dataset.

## A.2  Limiations

In this work, we demonstrate the superior high-resolution and multilingual text-to-image generation capabilities of Lumina-Next compared to open-source models like SDXL and PixArt-$\alpha$. However, our text-to-image generation model still fall short of state-of-the-art closed-source models like Midjourney and DALLE 3 in terms of text-image alignment and visual aesthetics. A notable gap lies in the size of text-image pairs used for multi-stage training. Although we expand the size of our training dataset to 20 million, it remains considerably smaller than the datsets used by these closed-source models. In addition, fine-tuning models with human preference data using techniques like Direct Preference Optimization is also important to improve image quality. We plan to address these challenges in future research.

---
**Algorithm 1** Few-step sampling using midpoint method and optimized schedule
---

> **procedure** MIDPOINTSAMPLER($\mathbf{v}_\theta(\mathbf{x}, t), t_{i \in \{0, \cdots, N\}}$)
>> **sample** $\mathbf{x}_0 \sim \mathcal{N}(0, \mathbf{I})$
>> **for** $i \in \{0, \cdots, N-1\}$ **do**
>>> $\Delta t \leftarrow t_{i+1} - t_i$             $\triangleright$ Adaptive step size
>>> $\mathbf{d}_i \leftarrow \mathbf{v}_\theta(\mathbf{x}_i, t_i)$          $\triangleright$ Evaluate $\mathrm{d}\mathbf{x}/\mathrm{d}t$ at $(\mathbf{x}, t_i)$
>>> $(\tilde{\mathbf{x}}_{i+1}, t_{i+1}) \leftarrow (\mathbf{x}_i + \frac{\Delta t}{2}\mathbf{d}_i, t + \frac{\Delta t}{2})$    $\triangleright$ Additional evaluation point
>>> $\mathbf{d}_{i+1} \leftarrow \mathbf{v}_\theta(\tilde{\mathbf{x}}_{i+1}, t_{i+1})$       $\triangleright$ Evaluate $\mathrm{d}\mathbf{x}/\mathrm{d}t$ at $(\tilde{\mathbf{x}}_{i+1}, t_{i+1})$
>>> $\mathbf{x}_{i+1} \leftarrow \mathbf{x}_i + \Delta t \mathbf{d}_{i+1}$       $\triangleright$ Second order step from $t_i$ to $t_{i+1}$
>> **end for**
>> **return** $\mathbf{x}_N$
> **end procedure**

---

# B  Additional Details Lumina-Next

## B.1  Illustration of Next-DiT's Architecture Improvements

In Figure 10, we illustrate the structural difference between Flag-DiT and Next-DiT. The main improvements include 3D RoPE, sandwich normalization, group-query attention, *etc*. Specifically, We leverage grouped-query attention [4] in transformer blocks, which is an interpolation between multi-head and multi-query attention to improve inference speed as well as reduce memory consumption. For instance, our 2B model divides 32 query heads into 8 groups, resulting in 8 shared key-value heads. This modification reduces the parameter size while achieving similar performance compared to the original multi-head attention.

## B.2  Sampling with Optimized Schedule and Higher-Order ODE Solver

In the main text, we propose two optimized time schedules and combine them with higher-order ODE solvers for few-step sampling. Algorithm B.1 illustrates the pseudocode for using the midpoint method with the optimized schedule.

## B.3  Improved Image-Text Pairs

Lumina-T2I constructs 14M high-aesthetic synthetic text-image pairs as the training dataset. While the limited data size leads to fast convergence, these data only cover a small proportion of real image distribution, which contains diverse contents and styles. Besides, the quality of text captions in this dataset is imbalanced, consisting of incomplete or inaccurate image descriptions. As pointed out in recent works [8, 14], improving the diversity and overall quality of text prompts can greatly boost text-image alignment. Therefore, it is important to re-examine the construction of training datasets.

**LAION Augmented High-Quality Images**  We first expand the size of our training dataset from 14M to 20M using an additional 6M text-image pairs from the LAION [69] dataset. To efficiently filter high-quality images from large-scale internet datasets, we employ a two-stage data filtering pipeline:

In the first stage, we aim to obtain a coarse-grained candidate pool from massive images. Specifically, we first filter out all images with resolutions lower than $512 \times 512$. Then, we further filter out images with low aesthetic scores ($< 5.5$), high watermark probabilities ($> 0.5$), and high unsafe probabilities ($> 0.5$). With these filtering thresholds, we construct an image pool with approximately 30 million items. In the second stage, we introduce the overall image quality rank as a fine-grained filtering metric to further obtain a subset with superior overall quality. Concretely, we rank the 30 million coarse-filtered images based on aesthetic score, watermark probability, and safety probability, respectively. Then we sum the three types of ranks into an overall image quality rank, and the top-ranking 6 million images are selected to form our final high-quality image dataset. Upon manual evaluation, the 6 million image data filtered by our two-stage pipeline exhibit high resolution, high aesthetic appeal, and minimal probabilities of watermarking and unsafe content.

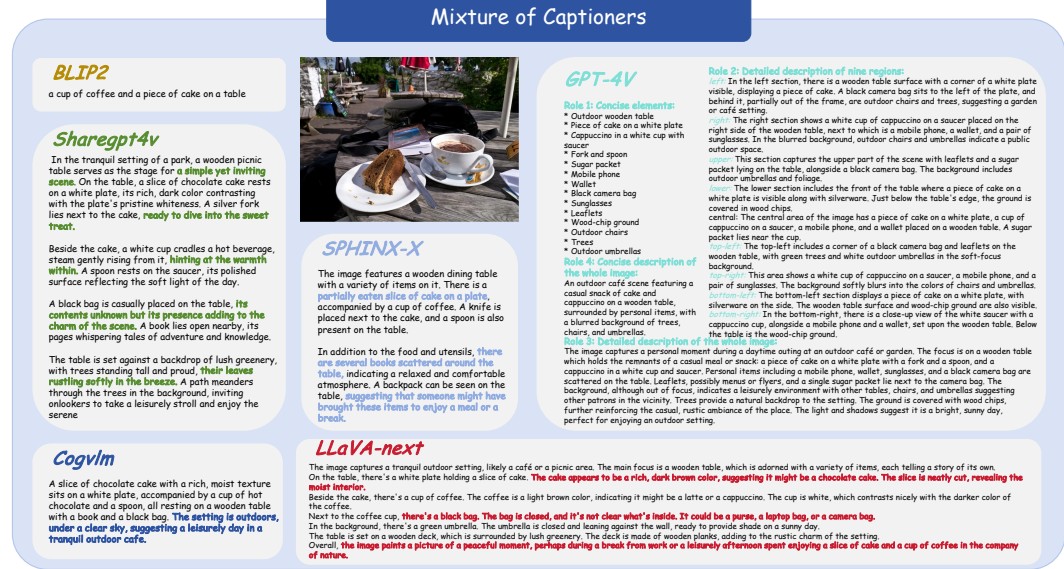

Figure 11: Illustration of mixture-of-captioners.

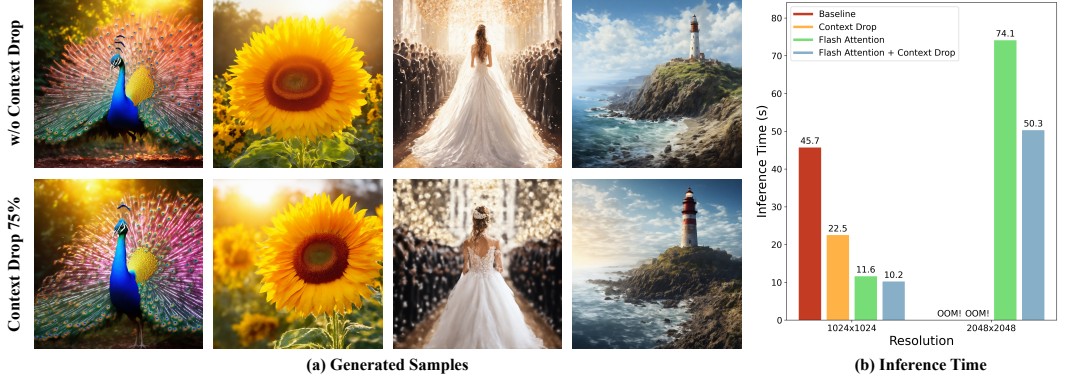

**(a) Generated Samples**                                    **(b) Inference Time**

Figure 12: (a) Qualitative results of 2K images generated by Lumina-Next with and without context drop. (b) Comparison of inference speed between different settings using 50 Euler steps.

**Mixture-of-Captioners**   Although PixArt-$\sigma$ [14] and DALLE-3 [8] highlight the importance of improving the quality of text descriptions, they only leverage a single captioner to enhance caption accuracy. As illustrated in Figure 11, we compare captions generated from open-sourced Vision-Language Models (VLMs) such as BLIP2 [44], LLaVa [48], SPHINX [34], and Share-GPT4V [17]. We observe different VLMs describe different information of the given image in different styles at different visual granularities. Moreover, these generated captions are complementary to each other since none of them achieves 100% captioning accuracy without any hallucination. Motivated by this observation, we propose a novel Mixture-of-Captioners (MoC) approach which employs multiple pretrained VLMs to generate mutual complementary captions. Besides, we collect a high-quality and high-resolution dataset consisting of 100K images and leverage GPT-4V to perform multi-facet holistic descriptions over these high-resolution images, depicted in Figure 11. After re-captioning using our proposed MoC, each image is paired with a caption pool with multiple descriptions from various models. During the training stage, we randomly sample one caption from collected MoC caption pools. This also resembles a kind of data augmentation that enhances model robustness during inference with all types of user inputs.

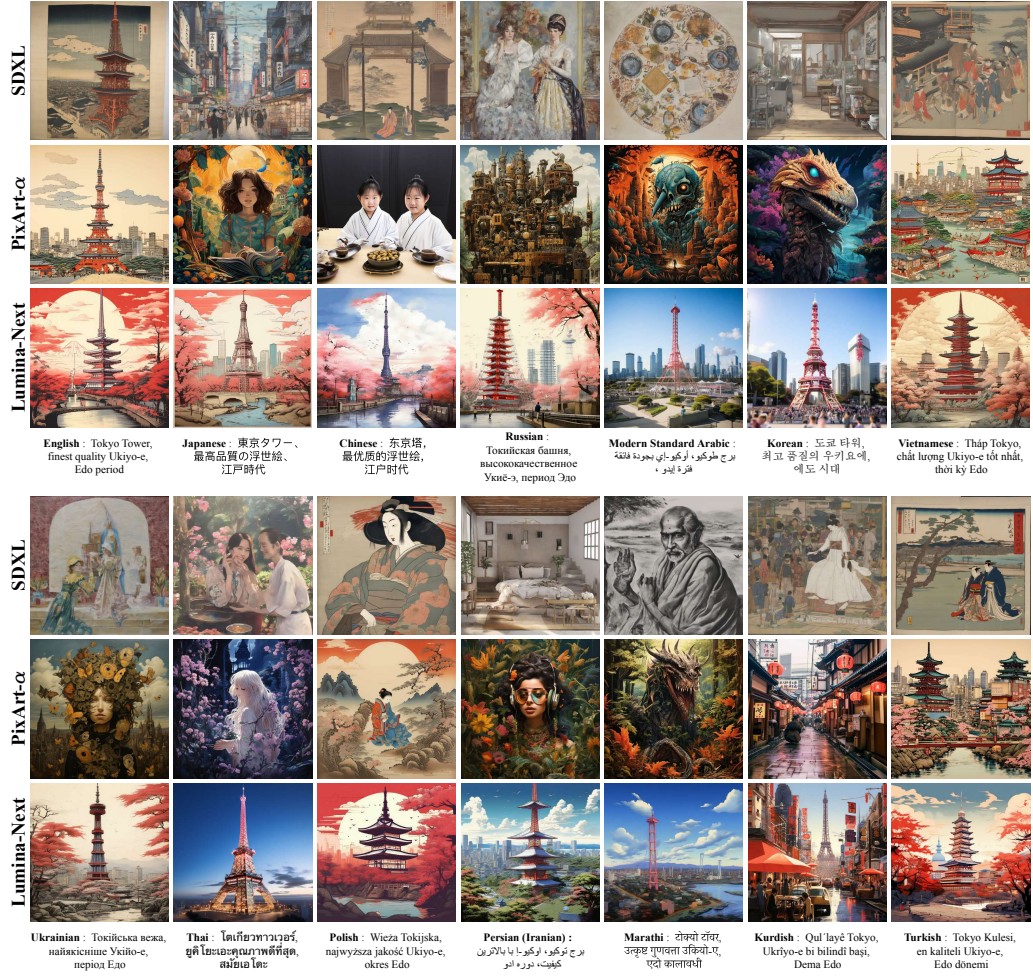

Figure 13: Results of multilingual text-to-image generation by Lumina-Next, SDXL, and PixArt-$\alpha$.

## C More Text to Image Generation Results

### C.1 Results of Time-Aware Context Drop

As shown in Figure 12 (a), Next-DiT using the proposed Time-Aware Context Drop with a 75% downsampling ratio generates comparable and even better 2K images compared to the baseline. The inference time comparison in Figure 12 (b) highlights that our Context Drop can further enhance inference speed when combined with advanced inference techniques for transformers, such as Flash Attention [22], which is more significant for higher-resolution image generation. Notably, our Context Drop is similar to masked attention or dilated convolution, which has been proven to be the key of resolution extrapolation in diffusion U-Net [37, 40]. Hence, this high-level resemblance in our diffusion transformer architecture may explain why Context Drop can reduce repetitive and unreasonable artifacts in high-resolution image generation, presenting an interesting avenue for future research.

### C.2 Results of Zero-shot Multilingual Generation

Previous T2I models, such as Imagen [67], PixArt-$\alpha$ [15] and Stable Diffusion [31, 65], utilized CLIP and T5 as text encoder for generating high aesthetics images. Such choices took advantage of pretrained text encoders or multimodal aligned text encoders as good representations for text-conditional image synthesis. However, employing T5 and CLIP as text embedding shows limited

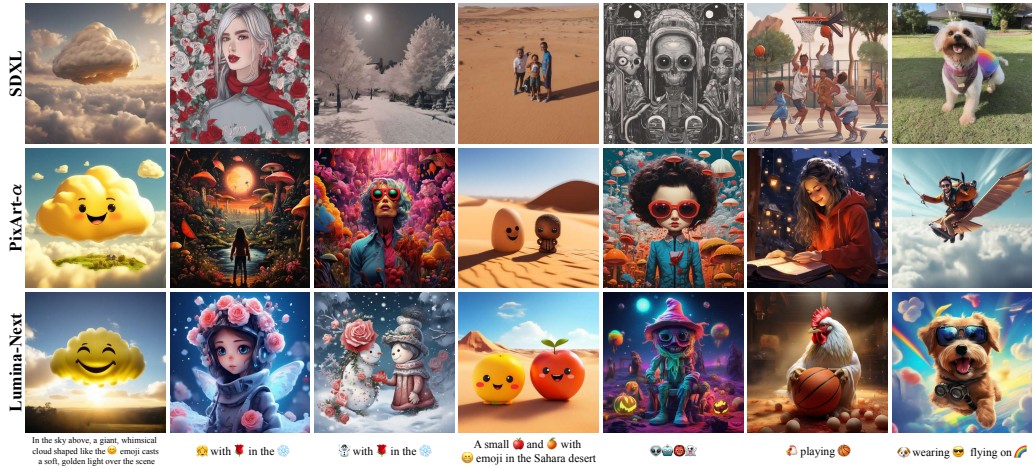

Figure 14: Results of text-to-image generation with emojis by Lumina-Next, SDXL, and PixArt-$\alpha$.

effectiveness in handling multilingual prompts. Different from those approaches, the Lumina series adopts decoder-only LLMs as text encoders. For instance, Lumina-T2X uses a 7 billion LLaMa as text encoder, while Lumina-Next employs the smaller Gemma-2B to decrease GPU memory costs and increase training throughput. Remarkably, both LLaMa-7B and Gemma-2B endow Lumina T2I models with the ability to understand multilingual prompts in a zero-shot manner, despite being primarily pretrained on English-only corpora. As shown in Figure 13, our T2I models not only accurately interpret multilingual prompts but also generate images with cultural nuances and show a preliminary ability to understand emojis, even though it was mainly trained with English-only image-text pairs. Compared to SDXL and PixArt-$\alpha$, which use CLIP and T5, there is a significant improvement in the understanding of multilingual prompts. We argue that decoder-only LLMs with strong language understanding abilities better align multilingual semantics than the encoder-decoder architecture of T5 or the multimodal alignment of CLIP text embeddings. To further explore the relationships between multilingual generation abilities and text encoders, we train Lumina-Next with different open-source LLMs, including Gemma-2B, Qwen-1.8B [5], and InternLM-7B [12]. As demonstrated in Figure 15, Lumina-Next with Qwen-1.8B and InternLM-7B showcase significantly better text-image alignment compared to Gemma-2B when using diverse Chinese prompts as inputs. Remarkably, Lumina-Next with Qwen-1.8B and InternLM-7B accurately captured the meanings and emotions of classic Chinese poems, reflecting these nuances in the generated images. This observation corresponds with the stronger multilingual abilities in Qwen-1.8B and InternLM-7B. Therefore, we argue that using more advanced LLMs as text encoders enhances text-to-image generation performance.

## C.3 Resolution Extrapolation with Any Aspect-ratio

In Figure 16, we further compare the panorama (1024 × 4096 images) generation performance of the proposed Lumina-Next with MultiDiffusion [7] and DemoFusion [29] to highlight its any aspect ratio extrapolation ability. We found that although panorama generation is one of the primary goals of MultiDiffusion, its "copy-paste" inference scheme easily leads to unreasonable content. DemoFusion adopts a progressive upscaling strategy, limiting its ability to generate arbitrary aspect ratios by the base model – in extreme aspect ratio scenarios, it also fails to produce satisfactory content. In contrast, Lumina-Next perfectly maintains the ability to generate at any aspect ratio under the training resolution, even when performing 4× resolution extrapolation.

## C.4 Generation with Long Prompt

Lumina-Next can handle image generation with long prompts. We give examples in Figure 17. As seen, with long prompts as inputs, Lumina-Next can produce high-quality images that strictly follow the details in the prompt. This ability remains when using other languages such as Chinese.

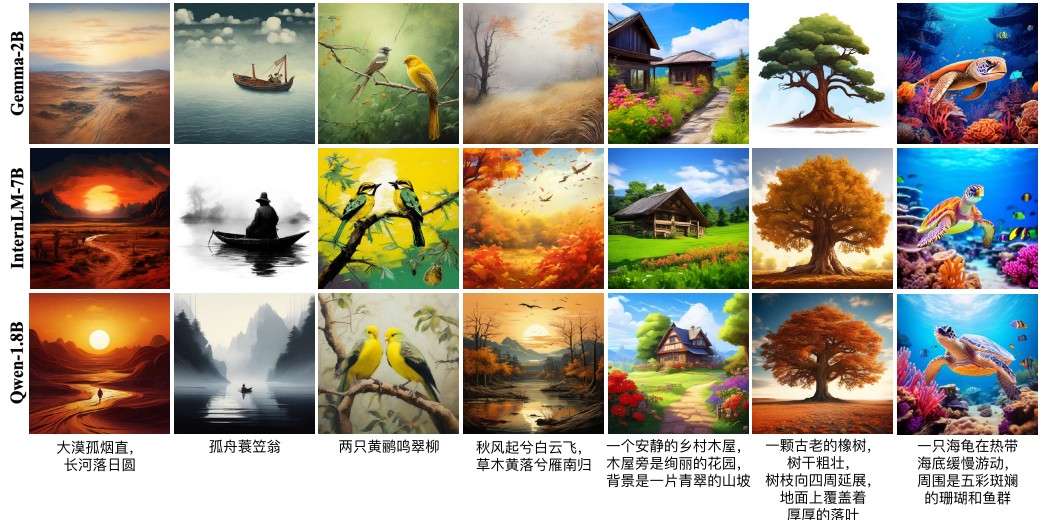

Figure 15: Results of multilingual text-to-image generation using different LLMs as text encoders.

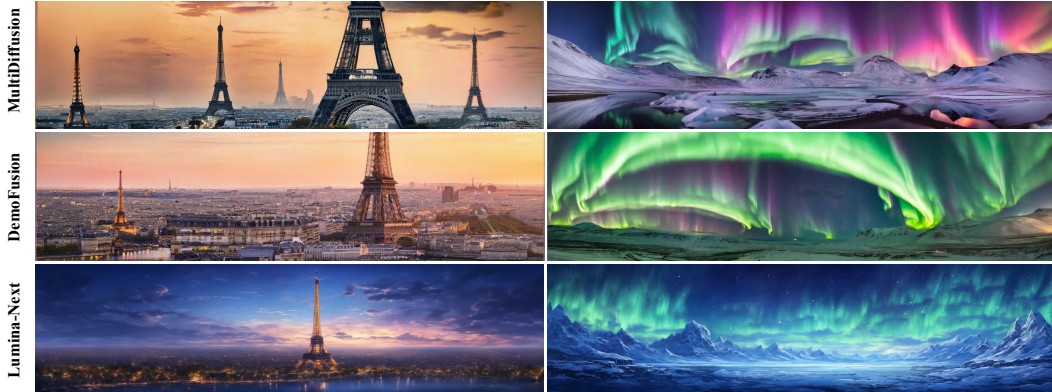

Figure 16: Results of $4\times$ resolution extrapolation with extreme aspect ratio.

# D Any Resolution Recognition with Lumina-Next

Lumina-Next is not only a generative modeling framework for generating images with various resolutions but also a framework for recognizing images at any resolution. As illustrated in Figure 18, our Next-DiT can be seamlessly adapted into a robust backbone for image representation learning. By incorporating the sandwich normalization and 2D RoPE design into our architecture, Next-DiT achieves superior performance and faster convergence while maintaining computational efficiency compared to the original vision transformer in image recognition tasks. Furthermore, Next-DiT overcomes the limitations of previous vision transformers in processing images of varying resolutions, demonstrating remarkable generalization capabilities to handle input images with arbitrary resolutions and aspect ratios.

## D.1 Pipelines

The unified framework of Lumina-Next for image generation and understanding at any resolution is depicted in Figure 18. Specifically, we introduce a learnable gate factor in the attention module, thereby avoiding the incorporation of time embeddings and label embeddings in image recognition tasks. We apply global average pooling (GAP) to all image tokens to obtain the class token, which is then fed into the classification head implemented by an MLP network. By integrating 2D RoPE during attention computation and incorporating RMSNorm before and after each attention and MLP layer, our Next-DiT architecture exhibits enhanced resolution extrapolation capabilities.

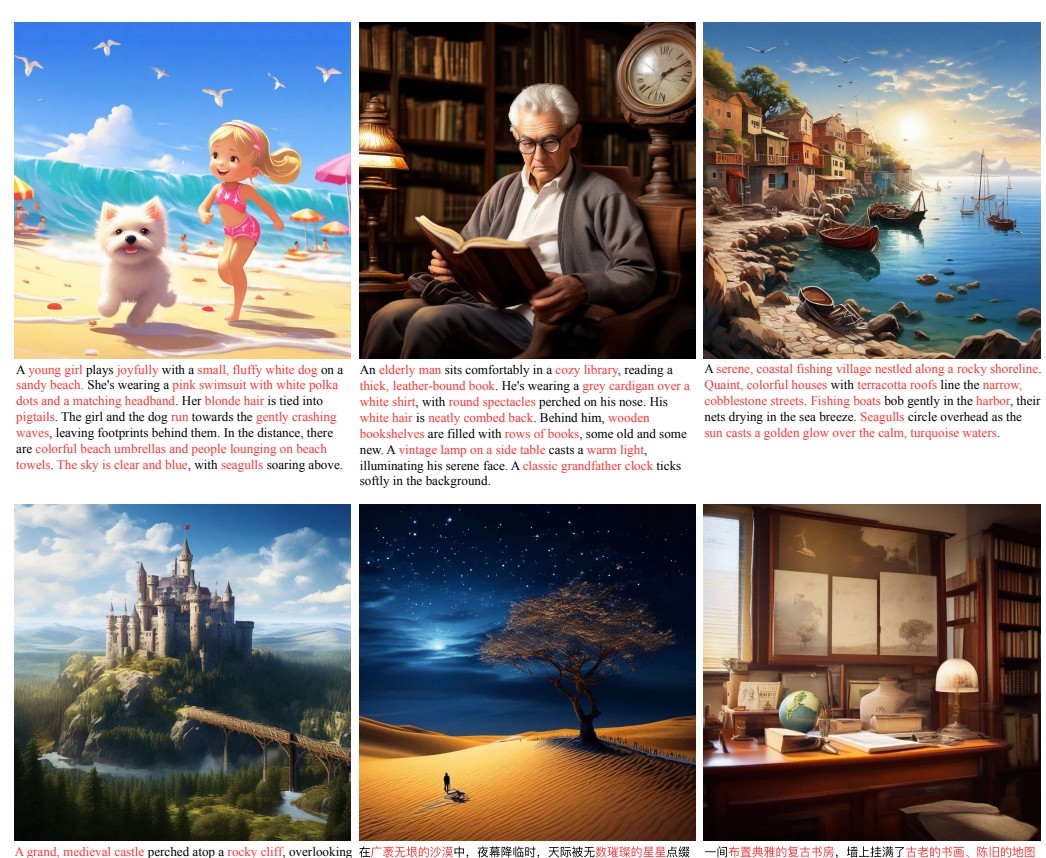

A young girl plays joyfully with a small, fluffy white dog on a sandy beach. She's wearing a pink swimsuit with white polka dots and a matching headband. Her blonde hair is tied into pigtails. The girl and the dog run towards the gently crashing waves, leaving footprints behind them. In the distance, there are colorful beach umbrellas and people lounging on beach towels. The sky is clear and blue, with seagulls soaring above.

An elderly man sits comfortably in a cozy library, reading a thick, leather-bound book. He's wearing a grey cardigan over a white shirt, with round spectacles perched on his nose. His white hair is neatly combed back. Behind him, wooden bookshelves are filled with rows of books, some old and some new. A vintage lamp on a side table casts a warm light, illuminating his serene face. A classic grandfather clock ticks softly in the background.

A serene, coastal fishing village nestled along a rocky shoreline. Quaint, colorful houses with terracotta roofs line the narrow, cobblestone streets. Fishing boats bob gently in the harbor, their nets drying in the sea breeze. Seagulls circle overhead as the sun casts a golden glow over the calm, turquoise waters.

A grand, medieval castle perched atop a rocky cliff, overlooking a vast, undulating landscape of forests and meadows. The castle's imposing stone towers and turrets rise majestically against a backdrop of a brilliant blue sky. A drawbridge spans a deep, mist-filled moat, and banners flap in the gentle breeze, displaying regal coats of arms.

在广袤无垠的沙漠中，夜幕降临时，天际被无数璀璨的星星点缀得如同一幅银色的华丽锦缎。月亮将温柔的月光洒在金黄色的沙丘上，投射出柔和的光影。顽强的树挺立在沙丘之间，静静地立在这寂静的夜色中。一个孤独的旅人缓缓行走，仿佛在追寻着星辰的指引。

一间布置典雅的复古书房，墙上挂满了古老的书画、陈旧的地图和稀有的收藏品。红木书架上摆放着各类珍贵的典籍，书脊泛着岁月的光泽。一张大书桌镇坐在房间中央，上面放着一本打开的古书，旁边是一支装饰精美的鹅毛笔和一个古董墨水瓶。昏黄的台灯发出柔和的光芒，映照在桌上的黄铜地球仪上。窗外的晚霞透过厚重的窗帘洒进屋内，给整个房间增添了一抹温暖的色彩。

Figure 17: Generated images of Lumina-Next with long prompts.

We employ a progressive training scheme with a fixed-resolution pre-training stage followed by an arbitrary-resolution fine-tuning stage, enabling Next-DiT to achieve robust image understanding across various resolutions.

**Fixed Resolution Pre-training** We first pre-train the Next-DiT architecture at a fixed resolution by cropping and resizing images to $224 \times 224$, following the approach of the original ViT. This resolution balances the trade-off between training speed and input resolution, as higher resolutions require more FLOPs and reduced throughput. After pre-training, we directly evaluate our model on unseen resolutions to guide a subsequent fine-tuning stage with arbitrary resolutions.

**Any Resolution Fine-tuning** To enable our model to handle images with any resolution and aspect ratio, we employ an any-resolution fine-tuning stage. While previous works [23, 78] also utilize a multi-resolution training approach, they rely on a set of predefined input resolutions and randomly sample different resolutions for each input image from this set. These methods often alter the original resolution of aspect ratio of the input images a lot, potentially causing a performance drop, and they incur additional computational costs by resizing low-resolution images to higher resolutions unnecessarily. In contrast, we propose a dynamic padding scheme that allows efficient model training while preserving the original resolution and aspect ratio of the input images. This approach enables us to fully exploit the multi-resolution understanding capability of Next-DiT in a computationally efficient manner.

**Dynamic Partitioning and Padding with Masked Attention** Our aim is to retain the original resolution and aspect ratio of each input image to the fullest extent during the subsequent fine-

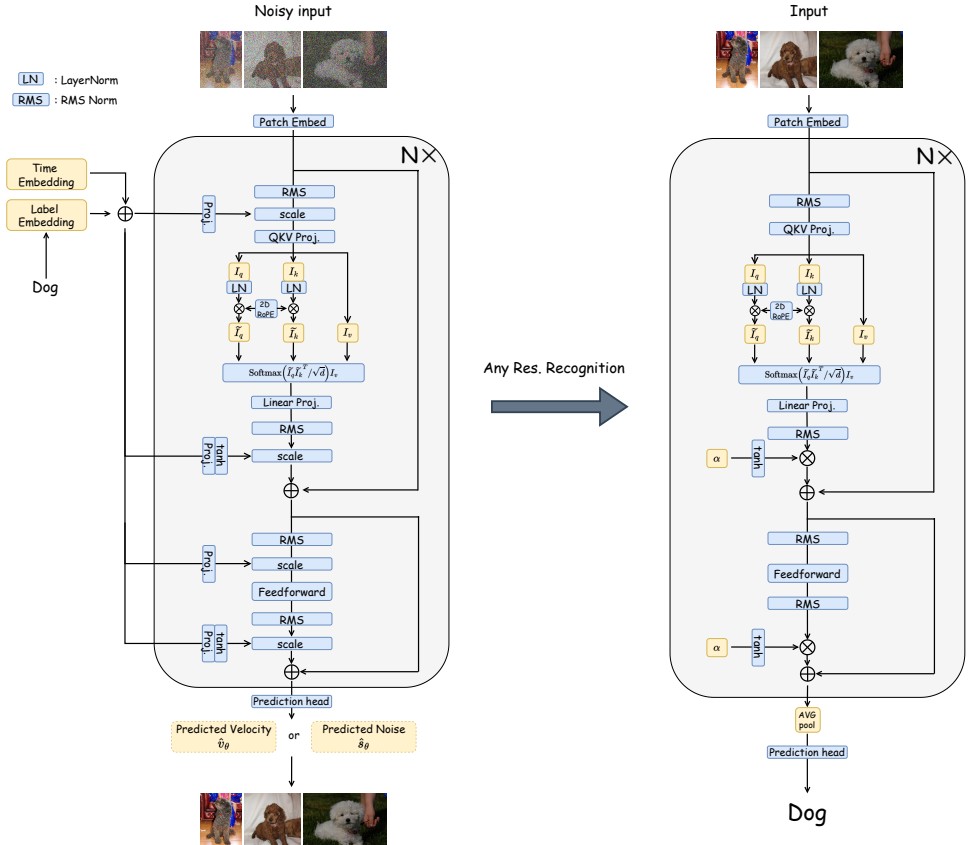

Figure 18: Comparison between the original architecture of Next-DiT and modified version for visual recognition.

tuning stage. Additionally, we seek to ensure that the width and height of the input image are divisible by the patch size of the patch embed layer, enabling us to fully utilize the image information during the embedding process. To achieve this, we propose a dynamic partitioning and padding scheme to optimize the handling of image tokens instead of relying on a fixed resolution, as illustrated in Figure 19. Specifically, given constraints on the maximum number of patches $N$ and the maximum aspect ratio $R_{\max}$, we define a set of candidate patch partitions $\mathcal{C} = \left\{ (H_p, W_p) \mid H_p \cdot W_p \leq N \text{ and } \frac{\max(H_p, W_p)}{\min(H_p, W_p)} \leq R_{\max} \right\}$. For an input image of size $(H_I, W_I)$, we determine its optimal patch partition by calculating the matching ratio between the original input size and the target size, given by $(H_p^*, W_p^*) = \mathrm{argmax}_{(H_p, W_p) \in \mathcal{C}} \frac{\min\left( \frac{H_p}{H_I}, \frac{W_p}{W_I} \right)}{\max\left( \frac{H_p}{H_I}, \frac{W_p}{W_I} \right)}$. Subsequently, we resize the input image to $(H_p^* \cdot P, W_p^* \cdot P)$, where $P$ denotes the patch size. This dynamic partitioning process leads to varying sequence lengths of patch tokens for each input image. To handle batches containing inputs with different token lengths, we pad all image token sequences to match the length of the longest sequence using a pad token. Additionally, we introduce attention masks within each attention module to prevent unwanted interactions between pad tokens and regular image tokens.

**Flexible Resolution Inference** By incorporating NTK-Aware Scaled RoPE and sandwich normalization, Next-DiT demonstrates exceptional resolution extrapolation during inference, even without fine-tuning on varied resolutions. Furthermore, the proposed dynamic partitioning and padding scheme allows for efficient fine-tuning while preserving image resolutions and aspect ratios. Consequently, our model can perform flexible inference on images with arbitrary resolutions and aspect ratios.

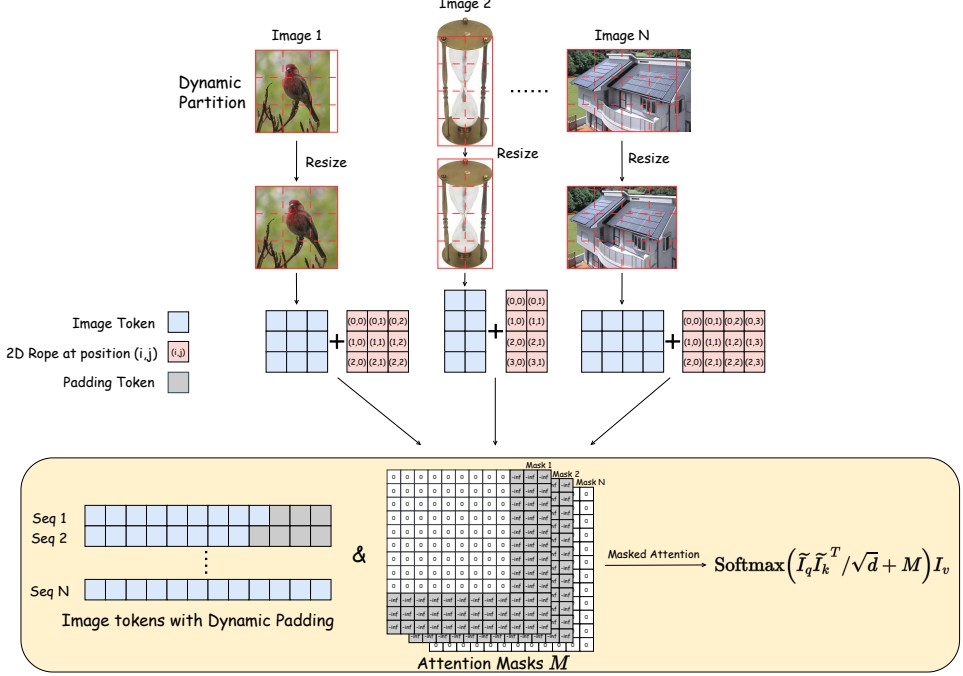

Figure 19: Illustration of our dynamic partitioning and padding scheme with masked attention for handling input images of arbitrary resolutions.

## D.2 Setups

We conduct experiments on the ImageNet-1K dataset to validate the effectiveness of Next-DiT for image recognition. We build our Next-DiT model following the architecture hyperparameters of ViT-base [28], stacking 12 transformer layers with a hidden size of 768 and 12 attention heads. This configuration ensures that our architecture has a comparable number of parameters to the original ViT. During the fixed-resolution pre-training stage, we train the models from scratch for 300 epochs with an input size of $224 \times 224$. We use the AdamW optimizer with a cosine decay learning rate scheduler, setting the initial learning rate, weight decay, and batch size to 1e-3, 0.05, and 1024, respectively. We follow the same training recipe as the DeiT model [79], including its data augmentation and regularization strategies. In the subsequent any-resolution fine-tuning stage, we further train the pre-trained model using the proposed dynamic partitioning and padding scheme. This fine-tuning is performed for an additional 30 epochs with a constant learning rate of 1e-5 and a weight decay of 1e-8.

## D.3 Experiments

We report the performance of our Next-DiT in Table 1. We first test our model at a fixed resolution of $224 \times 224$. Compared with DeiT-base, which has comparable parameters, Next-DiT exhibits superior performance and faster convergence. Notably, training the model for only 100 epochs achieves a Top-1 accuracy of 81.6%, nearly matching DeiT-base's result achieved after 300 epochs. Under the same training setting with 300 epochs of pre-training, Next-DiT surpasses DeiT-base with an accuracy of 82.3% versus 81.6%, respectively, demonstrating its effectiveness as a robust image backbone. We also adopt the flexible resolution inference strategy, preserving the original resolution and aspect ratio of the input images to validate the model's generalization capability across varied resolutions. After the additional fine-tuning stage, Next-DiT achieves 84.2% accuracy when handling arbitrary resolutions. We report the performance of DeiT-base pre-trained at $224 \times 224$, using the same inference configuration. To align with the input resolution, we interpolate its positional embeddings accordingly. Compared to our fine-tuned Next-DiT, the performance of DeiT-base drops significantly, suggesting its limitations in processing images with varied resolutions.

Table 1: Comparison of Next-DiT with DeiT [79] on ImageNet classification.

| Model | Params | Setting | Resolution | Top-1 Acc(%) |
|-------|--------|---------|------------|--------------|
| DeiT-base [79] | 86M | 300E | $224 \times 224$ | 81.8 |
| Next-DiT | 86M | 100E | $224 \times 224$ | 81.6 |
| Next-DiT | 86M | 300E | $224 \times 224$ | 82.3 |
| DeiT-base [79] | 86M | 300E | Flexible | 67.2 |
| Next-DiT | 86M | 300E+30E | Flexible | 84.2 |

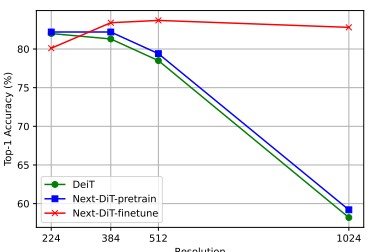

Figure 20: Performance of Next-DiT across different resolutions.

We also evaluate the performance across different image resolutions, as illustrated in Figure 20. Next-DiT demonstrates better generalization to larger image sizes compared to DeiT-base, even without fine-tuning. With additional fine-tuning using the proposed dynamic partitioning scheme, our model significantly enhances its ability to handle varied resolutions, particularly high resolutions such as $1024 \times 1024$, outperforming DeiT-base by a significant margin. These experimental results highlight the effectiveness of Next-DiT as a powerful image backbone for recognizing images at any resolution, in addition to its capability to generate images at various resolutions.

# E    Generating Multi-Views, Music and Audio with Lumina-Next

Lumina-Next is a versatile and expandable framework for generative modeling that goes beyond text-conditioning image generation. In this section, we demonstrate the application of Lumina-Next to multi-view generation using both image and text conditioning. Additionally, we extend Lumina-Next to encompass a wider range of modalities, including text-conditional audio, music, and point cloud generation.

## E.1    Image- and Text-conditional Multi-View Generation

**Pipelines**    Having witnessed the success of Next-DiT on text-to-image generation, we use Next-DiT's text-to-image model as a pre-trained model and extend it to image- and text-conditional multi-view generation, called MV-Next-DiT, as shown in Figure 21. Firstly, we introduce relative pose control to Next-DiT to distinguish between views. Specifically, for $N$ multi-view images with the same elevation and random azimuths rendering from 3D object, taking the azimuth of the first image as the reference view, the relative azimuths of the remaining $N - 1$ images to the first image are calculated sequentially. The obtained relative azimuths are encoded into the same dimension as time embedding using a 2-layer MLP network after $sin$ and $cos$ transformation, and are added to the time embedding to inject camera control into the model. Secondly, we expanded the original text condition of Next-DiT into text and image conditions. Specifically, the same encoding and injection method as Next-DiT is used for text condition. For the image condition, we use VAE to encode the input image, and then concatenate it to the multi-view noises ($N + 1$). This image is not used for supervision during training. In addition, when building the training data, this condition image has the same azimuth as the first supervised view (both are 0), but it is a random elevation. It is worth noting that the image condition is dropped based on probability during training, and the input of the image condition is optional during inference. Finally, a progressive training strategy of views and resolution is adopted during training, and views flexible inference strategy is used during inference. Specifically, during training, we first performed training at $256 \times 256$ resolution with $N = 4$. Then it was expanded to the training of $512 \times 512$ resolution with $N = 4$, and finally it was expanded to the training of $512 \times 512$ resolution with $N = 8$. In inference, since each object is trained using all random views, any number of view can be inferenced at one time. These improvements allow MV-Next-DiT to output multi-view images with high quality and flexible azimuth.

**Setups**    Data preparation: we use the Objaverse [24] list and caption provided by Cap3D [58] as training data. For data rendering, we first use the same elevation and 8 random azimuths for each object, render with the same camera setting, and obtain supervision data. Then the azimuths corresponding to 4 images are randomly selected from the 8 rendered views, and cooperate random

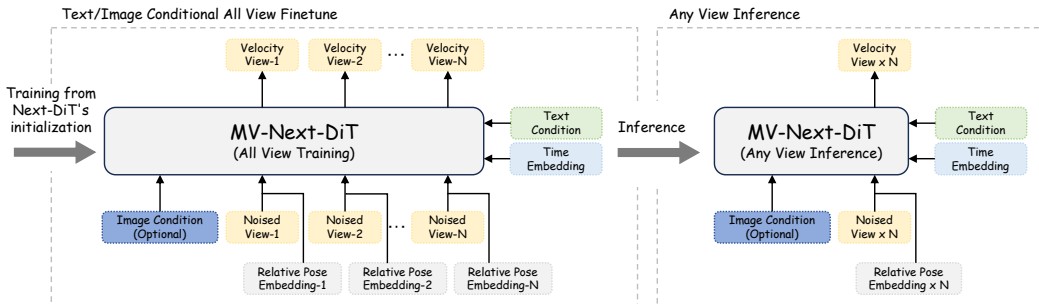

Figure 21: An illustration of multi-view images generation. Left: All view training paradigm of MV-Next-DiT. Right: Any view inference paradigm of MV-Next-DiT.

elevations to render input images. Each object then has 4 groups of training samples. Training details: We use Next-DiT with 600M parameters for training of MV-Next-DiT. The specific training settings are shown in Table 2.

Table 2: The detailed training setting of MV-Next-DiT.

| Training method | Pre-train model | Total batch (Images) | Learning rate | Iteration | A100 cost |
|---|---|---|---|---|---|
| $256 \times 256, N = 4$ | Next-DiT, Text-to-Image | $256 \times (N + 1)$ | 1e-4 | 100k | $16GPUs \times 45h$ |
| $512 \times 512, N = 4$ | MV-Next-DiT, $256 \times 256, N = 4$ | $128 \times (N + 1)$ | 1e-4 | 100k | $16GPUs \times 57h$ |
| $512 \times 512, N = 8$ | MV-Next-DiT, $512 \times 512, N = 4$ | $32 \times (N + 1)$ | 1e-4 | 100k | $16GPUs \times 72h$ |

**Experiments** We collected images provided by previous work, such as CRM [83], to test the performance of our MV-Next-DiT. In particular, we provide the generated results of only text condtion, text and image conditions, and the results of sampling 4 views, 6views, and 8views respectively from the setting of N=8, as shown in Figure 22. In addition, we compared the capabilities with existing multi-view methods as shown in Table 3. The overall performance demonstrates the advantages of our MV-Next-DiT. The rich higher-resolution flexible view generated results demonstrate MV-Next-DiTs excellent multi-view generation capabilities and the superiority of the Next-DiT architecture.

Table 3: Comparison of capabilities between MV-Next-DiT and other mutli-view methods.

| Methods | Base model | Resolution | Condition type | Inference views |
|---|---|---|---|---|
| Zero-123 [49] | SD Image Variations v2 | $256 \times 256$ | Image | 1 |
| MVDream [71] | SD v2.1 | $256 \times 256$ | Text | 4 |
| UniDream [51] | SD v2.1 | $256 \times 256$ | Text | 4 |
| ImageDream [82] | MVDream (SD v2.1) | $256 \times 256$ | Image | 4 |
| Wonder3D [52] | SD Image Variations v2 | $256 \times 256$ | Image | 6 |
| Zero-123++ [70] | SD v2 | $256 \times 256$ | Image | 6 |
| CRM [83] | ImageDream (SD v2.1) | $256 \times 256$ | Image | 6 |
| MV-Next-DiT | Next-DiT | $512 \times 512$ | Text & Image | $1 \sim 8$ |

### E.2 Text-Conditional Audio and Music Generation

**Pipelines** We use a modified audio VAE with a 1D-convolution-based model to derive audio latent. The audio signal is a sequence of mel-spectrogram sample $\mathbf{x} \in [0, 1]^{C_a \times T}$, where $C_a, T$ respectively denote the mel channels and the number of frames. Our spectrogram autoencoder is composed of 1) an encoder network $E$ which takes samples $\mathbf{x}$ as input and outputs latent representations $z$; 2) a decoder network $G$ reconstructs the mel-spectrogram signals $\mathbf{x}'$ from the compressed representation $\mathbf{z}$; and 3) a multi-window discriminator learns to distinguish the generated samples $G(\mathbf{z})$ from real ones in different multi-receptive fields of mel-spectrograms.

In text2audio generation, we include the dual text encoder architecture consisting of a main text encoder CLAP [30] that takes the original natural language caption $\mathbf{y}$ as input and a temporal encoder FLAN-T5 [20] which takes the structured caption $\mathbf{y}_s$ passed by LLM as input. The final conditional

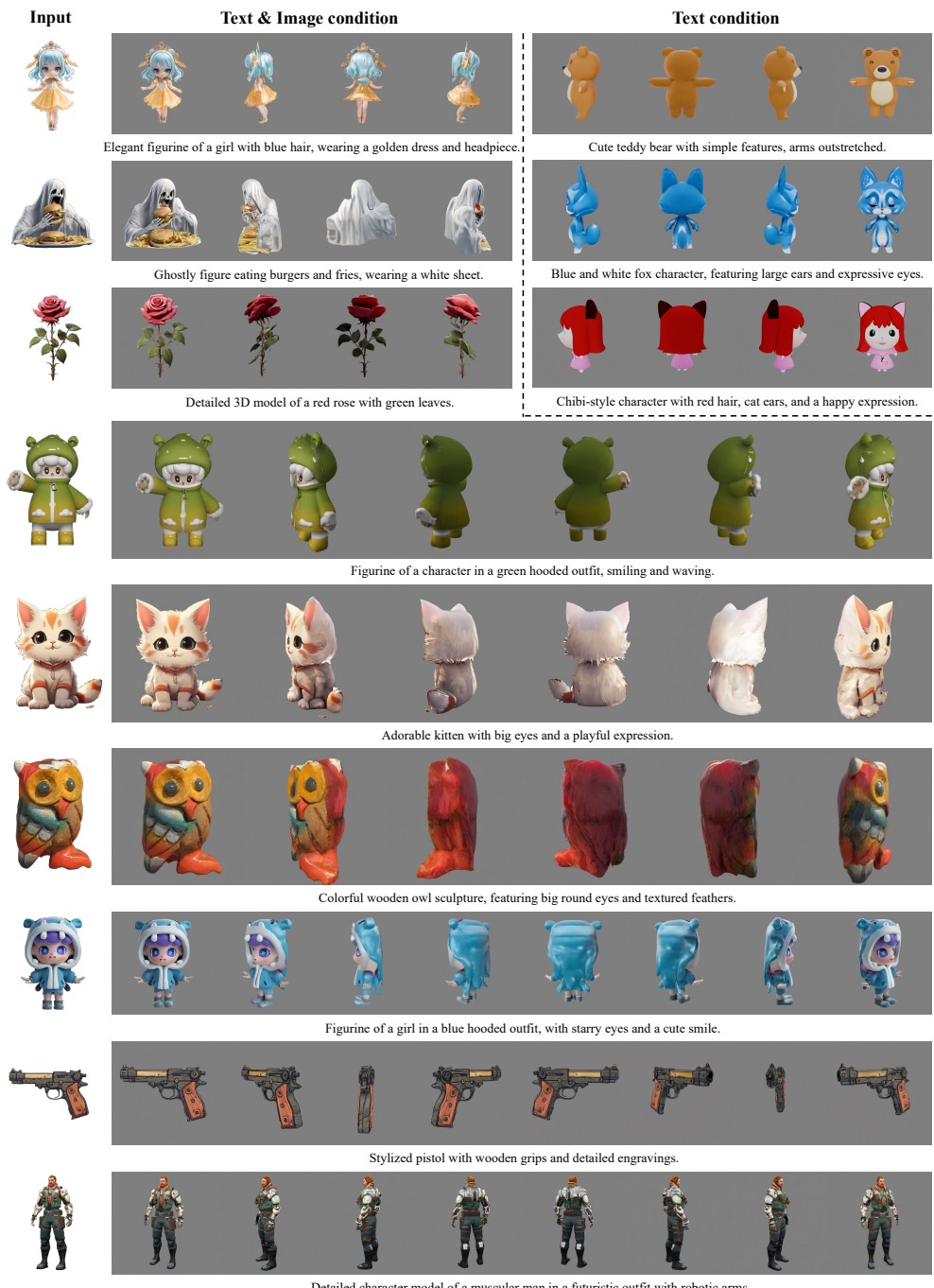

Figure 22: Results generated by MV-Next-DiT. The upper right is the result of text-to-multiview, and the rest is the result of text-&image-to-multiview, where the first column is the corresponding input image with background removed. In addition, every 3 rows from top to bottom are the generated results of 4 views, 6 views, and 8 views respectively.

representation is expressed as:

$$\mathbf{c} = \text{Linear}(\text{Concat}(f_{\text{text}}(\mathbf{y}), f_{\text{temp}}(\mathbf{y}_s))), \tag{5}$$

Where $f_{\text{text}}$ is the main text encoder and $f_{\text{temp}}$ is the temporal encoder. For the text encoder in text2music generation, we use the FLAN-T5 [20] text encoder that takes the original natural language caption $\mathbf{y}$ as input.

**Setups of Text-to-Music Generation**    We employ a diverse combination of datasets to facilitate the training process of our model. In the context of text-to-music synthesis, we exclusively employ the LP-MusicCaps [27] dataset. The culmination of these efforts yields a dataset comprising **0.92 million** audio-text pairs, boasting a cumulative duration of approximately **3.7K hours**.

We conduct preprocessing on both text and audio data as follows: 1) We convert the sampling rate of audios to 16kHz. Prior works [84, 41, 46] pad or truncate the audio to a fixed length (10s), while we group audio files with similar durations together to form batches to avoid excessive padding which could potentially impair model performance and slow down the training speed. This approach also allows for improved variable-length generation performance. We truncate any audio file that exceeds 20 seconds, to speed up the training process. 2) For audios without natural language annotation, we apply the pseudo prompt enhancement method from Make-An-Audio [41] to construct captions aligned with the audio. We traverse the AudioCaps training set and the LLM augmented data with a probability of 50%, while randomly selecting data from all other sources with a probability of 50%. For the latter dataset, we use "<text & all>" as their structured caption.

**Experiments of Text-to-Music Generation**    In this part, we compare the generated audio samples with other systems, including 1) GT, the ground-truth audio; 2) MusicGen [21]; 3) MusicLM [3]; 4) Mousai [68]; 5) Riffusion [33]; 6) MusicLDM [16]; 7) AudioLDM 2 [47]. The results are presented in Table 4, and we have the following observations: Regarding audio quality, our model consistently surpasses diffusion-based methods and language models across FAD and KL metrics. In terms of subjective evaluation, our model demonstrates the strong text-music alignment faithfulness.

Table 4: The comparison with baseline models on the MusicCaps Evaluation set. We borrow the results of Mousai, Melody, and MusicLM from MusicGen [21].

| Model | Objective Metrics | | Subjective Metrics | |
|---|---|---|---|---|
| | FAD ($\downarrow$) | KL ($\downarrow$) | MOS-Q($\uparrow$) | MOS-F($\uparrow$) |
| GroundTruth | / | / | 88.42 | 90.34 |
| Riffusion | 13.31 | 2.10 | 76.11 | 77.35 |
| Mousai | 7.50 | / | / | / |
| Melody | 5.41 | / | / | / |
| MusicLM | 4.00 | / | / | / |
| MusicGen | 4.50 | 1.41 | 80.74 | 83.70 |
| MusicLDM | 5.20 | 1.47 | 80.51 | 82.35 |
| AudioLDM 2 | 3.81 | 1.22 | 82.24 | 84.35 |
| **Ours** | **3.75** | **1.24** | **83.56** | **85.69** |

**Setups of Text-to-Audio Generation**    Following the benchmark studies [43, 41], we train on AudioSet-SL [46] and finetune the model in the Audiocaps dataset. Overall, we have ~3k hours with 1M audio-text pairs for training data. For evaluating text-to-audio models, the AudioCaption validation set is adopted as the standard benchmark, which contains 494 samples with five human-annotated captions in each audio clip.

We conduct preprocessing on the text and audio data: 1) convert the sampling rate of audios to 16kHz and pad short clips to 10-second long; 2) extract the spectrogram with the FFT size of 1024, hop size of 256 and crop it to a mel-spectrogram of size $80 \times 624$; 3) non-standard words (e.g., abbreviations, numbers, and currency expressions) and semiotic classes [77] (text tokens that represent particular entities that are semantically constrained, such as measure phrases, addresses, and dates) are normalized.

Our models conduct a comprehensive evaluation using both objective and subjective metrics to measure audio quality, text-audio alignment fidelity, and inference speed. Objective assessment includes Kullback-Leibler (KL) divergence, Frechet audio distance (FAD), and CLAP score to quantify audio quality. In terms of subjective evaluation, we conduct crowd-sourced human assessments employing the Mean Opinion Score (MOS) to evaluate both audio quality (MOS-Q) and text-audio alignment faithfulness (MOS-F).

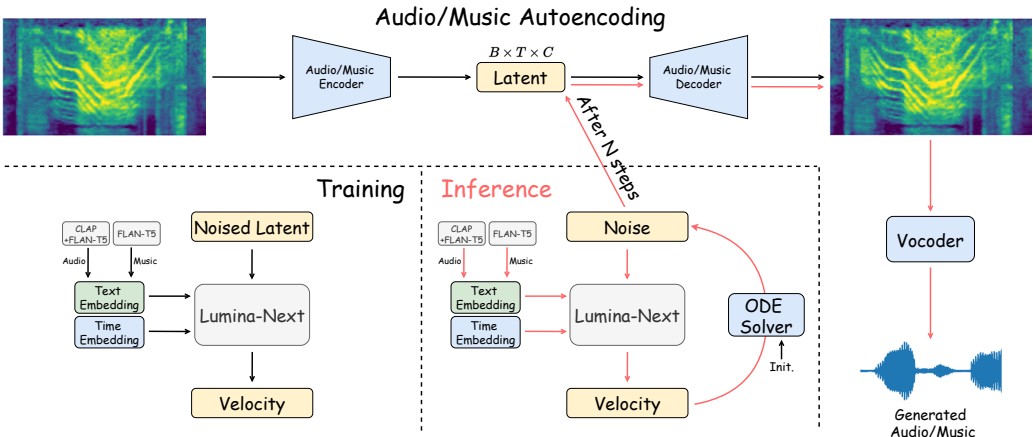

Figure 23: An illustration of text-guided music/audio generation. It consists of the following main components: 1) VAE to encode spectrogram into a latent and convert it back to spectrogram; 2) text encoder to derive high-level textual representation; 3) flow transformer to inject condition; and 4) separately-trained neural vocoder to convert mel-spectrograms to raw waveforms. In the following sections, we describe these components in detail.

**Experiments of Text-to-Audio Generation**  We conduct a comparative analysis of the quality of generated audio samples and inference latency across various systems, including GT (i.e., ground-truth audio), AudioGen [43], Make-An-Audio [41], AudioLDM-L [47], TANGO [36], Make-An-Audio 2 [39], and AudioLDM 2 [47], utilizing the published models as per the respective paper and the same inference steps of 100 for a fair comparison. The evaluations are conducted using the AudioCaps test set and then calculate the objective and subjective metrics. The results are compiled and presented in Table 6. From these findings, we draw the following conclusion:

In terms of audio quality, our proposed system demonstrates outstanding performance with the FAD of 1.03, demonstrating minimal spectral and distributional discrepancies between the generated audio and ground truth. Human evaluation results further confirm the superiority, with MOS-Q and MOS-F scores of 77.53 and 76.52, respectively. These findings suggest a preference among evaluators for the naturalness and faithfulness of audio synthesized by our model over baseline approaches.

For ablation, we 1) explore the effectiveness of dual text encoders for a deep understanding of arbitrary natural language input and observe that the drop in CLAP encoders has witnessed the distinct degradation of text-audio alignment faithfulness; 2) study the usage of Audioset data, and find that without fine-tuning on Audiocaps following previous works [47], a significant drop on objective evaluation happens.

Table 5: The audio quality comparisons with baselines.

| Model | Objective Metrics | | | Subjective Metrics | |
|---|---|---|---|---|---|
| | FAD ($\downarrow$) | KL ($\downarrow$) | CLAP ($\uparrow$) | MOS-Q($\uparrow$) | MOS-F($\uparrow$) |
| GT | / | / | 0.670 | 86.65 | 84.23 |
| AudioGen-Large | 1.74 | 1.43 | 0.601 | / | / |
| Make-An-Audio | 2.45 | 1.59 | 0.616 | 70.32 | 66.24 |
| AudioLDM | 4.40 | 2.01 | 0.610 | 64.21 | 60.96 |
| Tango | 1.87 | 1.37 | 0.650 | 74.35 | 72.86 |
| AudioLDM 2 | 1.90 | 1.48 | 0.622 | / | / |
| AudioLDM 2-Large | 1.75 | 1.33 | **0.652** | 75.86 | 73.75 |
| Make-An-Audio 2 | 1.80 | **1.32** | 0.645 | 75.31 | 73.44 |
| Ours | **1.03** | 1.45 | 0.630 | **77.53** | **76.52** |
| Ours (w/o Dual-encoder) | 1.48 | 1.53 | 0.601 | 75.13 | 72.01 |
| Ours (w/ Audioset) | 2.09 | 2.06 | 0.562 | / | / |

Table 6: Ablation studies. We use DiT to denote the same model architecture but with DDPM formulation.

| Model | FAD ($\downarrow$) | KL ($\downarrow$) | CLAP ($\uparrow$) |
|---|---|---|---|
| Next-DiT | **1.03** | 1.45 | 0.630 |
| w/o Dual-encoder | 1.48 | 1.53 | 0.601 |
| w/ Audioset | 2.09 | 2.06 | 0.562 |
| DiT | 1.56 | 1.91 | 0.573 |

Table 7: Quantitative results of point cloud generation. We multiplied the value of CD by $10^3$.

| Shape | Model | MMD ($\downarrow$) | COV (%, $\uparrow$) | Shape | Model | MMD ($\downarrow$) | COV (%, $\uparrow$) |
|---|---|---|---|---|---|---|---|
| | PC-GAN | 3.819 | 42.17 | | PC-GAN | 13.436 | 46.23 |
| | TreeGAN | 4.323 | 39.37 | | TreeGAN | 14.936 | 38.02 |
| | PointFlow | 3.688 | 44.98 | | PointFlow | 13.631 | 41.86 |
| Airplane | ShapeGF | 3.306 | 50.41 | Chair | ShapeGF | 13.175 | **48.53** |
| | PDiffusion | **3.276** | **48.71** | | PDiffusion | **12.276** | 48.94 |
| | Ours | 3.371 | 49.21 | | Ours | 12.975 | 48.33 |

## E.3 Label- and Text-Conditional Point Cloud Generation

**Pipelines** We propose a density-invariant point cloud generator to validate the effectiveness of Lumina-Next in the 3D domain. The density-invariant property allows an efficient *training on a fixed point number* (*e.g.*, 256 points) and an *inference of any point number* (*e.g.*, 1024, 2048, or more points). Different from 2D images, increasing the density of a point cloud does not affect the overall shape but rather enhances the details. To achieve any point inference, we combine the Time-aware Scaled RoPE with Fourier features [76] and design a Time-aware Scaled Fourier feature.

$[\cos(2\pi\boldsymbol{\theta}_1^\top \mathbf{p}), \sin(2\pi\boldsymbol{\theta}_1^\top \mathbf{p}), ..., \cos(2\pi\boldsymbol{\theta}_d^\top \mathbf{p}), \sin(2\pi\boldsymbol{\theta}_d^\top \mathbf{p}), ...]$. To inject the time condition of diffusion steps into the Fourier feature, we scale the frequency via $\boldsymbol{\theta}'_d = \boldsymbol{\theta}_d \cdot s^{\frac{d_{\text{head}}}{d_t}}$, where $d_t = (d_{\text{head}} - 1) \cdot t + 1$. This leads the generator to focus on the global information in the early steps of denoising, and include more shape details in later steps. This Time-aware Scaled Fourier feature can effectively distinguish continual positions in 3D space, thus our model guarantees any point inference even though the model is only trained on a fixed low density.

**Setups** We train a label and a textual conditioned 3D generator respectively. For the label conditional model, we employ the ShapeNet dataset [13] and follow [56] to preprocess it. We randomly sample 256 points from each point cloud to train the model. After training, we synthesize point clouds of densities in $\{256, 512, 1024, 2048, 4096, 8192\}$. To evaluate the generation, we adopt the Minimum Matching Distance (MMD) and the Coverage score (COV) as quality metrics and use Chamfer Distance (CD) as the distance measure to calculate them. For the textual guided generator, we adopt the Cap3D dataset [57] to train the model, where we use RoPE positional embedding instead of the time-aware Fourier feature.

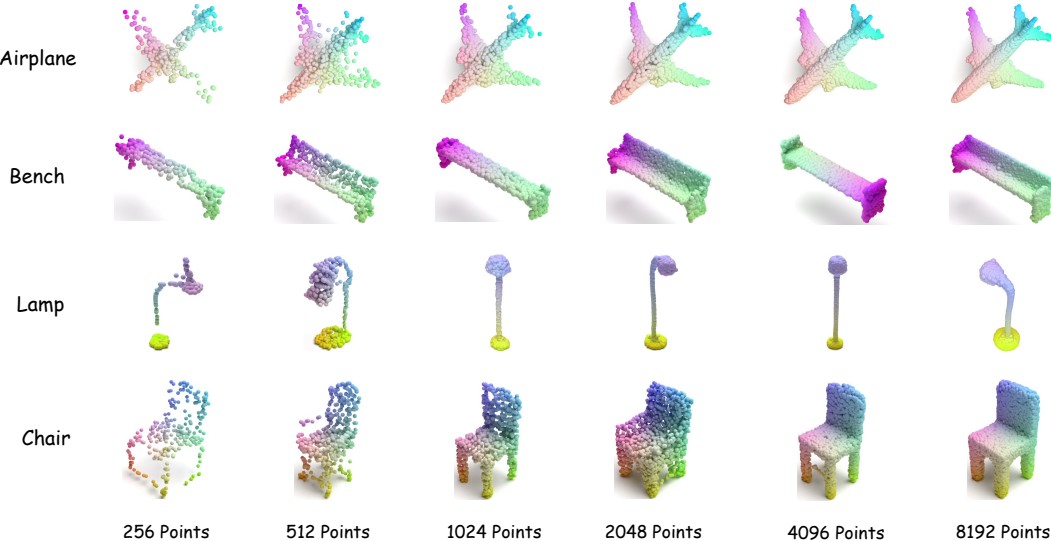

Figure 24: Examples of generated point clouds with different densities, sampled from the generator trained on 256 points.

**Experiments**   We compare the generation performance of two categories: airplane and chair. We fine-tune our pre-trained generator on 2048 points to further improve the generation quality. We mainly compare our performance with PC-GAN [2], TreeGAN [72], PointFlow [85], ShapeGF [11], and PDiffusion [56] in Table 7. We observe that our generator achieves a satisfactory performance compared to existing models. We also visualize some generated point clouds in Figure 24, where we directly apply the generator pre-trained on 256 points to sample higher-density point clouds. We observe the density of generated point clouds decreases when we increase the density (*e.g.*, airplane and chair), which could be due to the fact that more points would cause a smoother attention matrix in self-attention layers.

