# OpenReview forum: "Lumina-Next : Making Lumina-T2X Stronger and Faster with Next-DiT"
_NeurIPS.cc/2024/Conference — NeurIPS 2024 poster_

### Official Review · Reviewer_Nfk5 · 2024-07-12

**Soundness:** 4
**Presentation:** 4
**Contribution:** 3
**Rating:** 6
**Confidence:** 4

**Summary:**

Lumina-Next is an improved version of Lumina-T2X, featuring a core architecture  that employs a Flow-based Large Diffusion Transformer (Flag-DiT). Through empirical experiments and analysis, Lumina-Next introduces an enhanced Next-Dit architecture and develops a fast sampling algorithm, which boosts the model's generalization capability and improves both training and inference efficiency.

**Strengths:**

This paper focuses on Lumina-T2X and provides empirical analysis and discussion on the training stability, inference efficiency, and extrapolation performance of this type of Flow-based Large Diffusion Transformer. Its advantages are as follows:

- This paper uses 3D RoPE to replace 1D RoPE, enabling the perception of positional information in both spatial and temporal dimensions without the need for learnable [nextline] and [nextframe] identifiers. These improvements help stabilize training and benefit the extrapolation of visual sequence context.
- Sandwich normalization is applied to the model structure to prevent the accumulation of errors due to increased model depth, thereby preserving model performance.
- Different extrapolation schemes is ablated, and Frequency- and Time-Awareness Scaled RoPE was proposed to improve resolution extrapolation.
- A time schedule suitable for flow-based diffusion models was proposed, improving the visual quality of fast sampling produced with 10-20 NFEs.

Overall, this paper addresses some of the shortcomings of Lumina-T2X and provides insights for the development of large-scale flow-based diffusion transformers.

**Weaknesses:**

- Whether identifiers such as [nextline] and [nextframe] are unnecessary when using 3D RoPE is not sufficiently discussed in the paper.
- The paper lacks a comparison with stable-diffusion-3, which is also a flow-based diffusion model.
- Is the proposed Time Schedule universally applicable at low NFEs (<10)? Figure 7 contains too few examples to reflect the actual effect. Are there any numerical comparisons?

Overall, this paper leans towards empirical research and analysis, making it more akin to a technical report.

**Questions:**

see weaknesses.

**Limitations:**

Although the paper discusses limitations in the supplementary materials, I believe it could benefit from adding some discussion on whether the training data might introduce biases related to violence, pornography, race, etc.

---

> ### Author Rebuttal · Authors · 2024-08-07
>
> ### Q1: Whether identifiers such as [nextline] and [nextframe] are unnecessary when using 3D RoPE is not sufficiently discussed in the paper.
>
> Good point! Lumina-T2X adopted learnable [nextline] and [nextframe] tokens to achieve flexible modeling of 2D/3D signals with 1D RoPE. However, we found that modeling any modality as the 1D sequence leads to certain limitations, e.g., misalignment with the image/video's physical properties and limited multi-dimensional resolution extrapolation ability (refer to Q1@Reviewer NAeC for more details). For these reasons, we introduced 2D/3D RoPE in lumina-next to replace 1D RoPE with learnable tokens. In this case, we naturally don't need learnable tokens to assist the model in understanding the 2D / 3D structures. We shall supplement the relevant discussion upon acceptance.
>
> ### Q2: The paper lacks a comparison with Stable Diffusion 3, which is also a flow-based diffusion model.
>
> Thanks for the suggestion! The SD3 model remains closed-source when submitting this paper. Here we further supplement the quantitative comparison results with SD3. To better illustrate the aesthetics and semantic ability, we conduct an AI preference study to evaluate Lumina-Next against other text-to-image models, since conventional metrics such as FID and CLIP-Score may not accurately reflect the generation quality. Following PixArt, we employ GPT-4o, the SoTA multimodal LLM exhibiting strong alignment with human preference, as our evaluator to vote based on image quality. As shown in the following table, Lumina-Next demonstrates competitive performance with advanced text-to-image models including PixArt and SD3. Note that SD3 uses over 1B text-image pairs, which is ~100x greater than our models, PixArt, which is already a training-efficiencient model, still uses 3x training compute than ours. However, we have to admit that Lumina-Next still underperforms these SoTA models in terms of text-image alignment and compositional generation, due to inadequate data and training.
>
> | Model | Winrate |
> | --- | --- |
> | SD3 | 69.5% |
> | PixArt | 43.6% |
>
> ### Q3: Is the proposed Time Schedule universally applicable at low NFEs (<10)? Figure 7 contains too few examples to reflect the actual effect. Are there any numerical comparisons?
>
> Thanks for posing this question. As a zero-shot method simply changing the time schedule, it is hard to get satisfactory results using extremely low NFEs (<10). Usually, using ~20 NFEs can get decent results. Besides, the design of the temporal condition in Time-aware scaled RoPE makes it intuitively more sensitive to the inference step number. Here, we give the results for NFEs of 40, 20, and 10 with NTK/Time- aware Scaled RoPE, and we can see that resolution extrapolation with Time-aware scaled RoPE is getting worse. However, the overall trend is consistent with the degradation of image quality in the case of using NTK-aware Scaled RoPE, and there is no significant nullification. We will add these experimental results to the main paper along with the visualization results.
>
> | Method | NFE | Time-aware Scaled RoPE | NTK-aware Scaled RoPE |
> | --- | --- | --- | --- |
> | Time-aware Scaled RoPE | 40 | 28.93 | 58.66 |
> | Time-aware Scaled RoPE | 20 | 28.34 | 60.07 |
> | Time-aware Scaled RoPE | 10 | 28.21 | 67.98 |
> | NTK-aware Scaled RoPE | 40 | 28.21 | 61.89 |
> | NTK-aware Scaled RoPE | 20 | 27.82 | 62.19 |
> | NTK-aware Scaled RoPE | 10 | 27.71 | 68.59 |

---

> ### Comment · Reviewer_Nfk5 · 2024-08-12
>
> Thank you for taking the time to respond.
> For Q3, what do the two columns of metrics represent respectively?

---

> ### Author Response · Authors · 2024-08-12
>
> Thanks for posing this question and sorry for our typos. The columns represent CLIP Score and FID, respectively.

---

> > ### Comment · Reviewer_Nfk5 · 2024-08-13
> >
> > Thank you for taking the time to respond.
> >
> > I see. I think you need to state in your paper that relatively good results can be achieved with NFE of 5-10, rather than achieving high-quality text-to-image generation samples within only 5 to 10 steps as you describe, which is suspected of misleading the reader and overl-claiming.

---

> > > ### Author Response · Authors · 2024-08-13
> > >
> > > We greatly appreciate your response and valuable suggestions, which will improve the quality and impact of our paper. We will incorporate your feedback and update our paper accordingly in the revision.

---

### Official Review · Reviewer_XpYM · 2024-07-12

**Soundness:** 3
**Presentation:** 3
**Contribution:** 3
**Rating:** 6
**Confidence:** 3

**Summary:**

The paper introduces Lumina-Next, an enhanced version of the Lumina-T2X model, which is a Flow-based Large Diffusion Transformer (Flag-DiT) aimed at transforming noise into various modalities like images and videos based on text instructions. Compared with Lumina-T2X, Lumina-Next introduces the following improvements:
* A redesigned architecture named Next-DiT, featuring 3D Rotary Position Embedding (RoPE) and sandwich normalization to stabilize training and inference.
* Frequency- and Time-Aware Scaled RoPE for better resolution extrapolation in text-to-image generation.
* An optimized time discretization schedule combined with higher-order Ordinary Differential Equation (ODE) solvers for faster and high-quality generation.

The model demonstrates stronger performance in text-to-image generation, resolution extrapolation, and multilingual capabilities in a zero-shot manner. It also shows versatility across different tasks such as visual recognition, multi-views, audio, music, and point cloud generation.

**Strengths:**

- This paper, like SD3 and PixArt, explores image generation within the DiT architecture, and the codes and models have been open-sourced, which is beneficial to the community.
- Strong functionality. Without additional training, Lumina-Next can easily generate higher-resolution images and produce images of decent quality in fewer inference steps.

**Weaknesses:**

- The paper mainly demonstrates the advantages over comparative methods in terms of functionality but does not show comparisons with these state-of-the-art methods in terms of text-image alignment and image quality.
- Although the paper's method is about DIT, it does not demonstrate the manifestation and properties of scaling laws in the generation domain.

**Questions:**

na

**Limitations:**

yes

---

> ### Author Rebuttal · Authors · 2024-08-07
>
> ### Q1: The work does not show comparisons with these state-of-the-art methods in terms of text-image alignment and image quality.
>
> Thank you for the suggestion. We have further supplemented the quantitative comparison experiments. Due to time constraints, we only compared Lumina-Next with representative SoTA T2I models, SD3 and PixArt. To better illustrate the aesthetics and semantic ability, we conduct an AI preference study to evaluate Lumina-Next against other text-to-image models, since conventional metrics such as FID and CLIP-Score may not accurately reflect the generation quality. Following PixArt, we employ GPT-4o, the SoTA multimodal LLM exhibiting strong alignment with human preference, as our evaluator to vote based on image quality. As shown in the following table, Lumina-Next demonstrates competitive performance with advanced text-to-image models including PixArt and SD3. Note that SD3 uses over 1B text-image pairs, which is ~100x greater than our models, PixArt, which is already a training-efficiencient model, still uses 3x training compute than ours. However, we have to admit that Lumina-Next still underperforms these SoTA models in terms of text-image alignment and compositional generation, due to inadequate data and training.
>
> | Model | Winrate |
> | --- | --- |
> | SD3 | 69.5% |
> | PixArt | 43.6% |
>
> ### Q2: The paper does not demonstrate the manifestation and properties of scaling laws in the generation domain.
>
> Good point! The nature of good scaling up and the consequent emergence of modeling capabilities are essential properties for evaluating GenAI nowadays, which has been explored in Lumina-T2X -- scaling up DiT from 0.6B to 7B. However, in this paper, since the scale of data we use is much smaller than the leading open-source models (e.g., 30M in Lumina-Next V.S. over 1B in SD3), the benefits of further scaling up the model to more parameters are not significant. Therefore, we adopt a 2B model to achieve a sweet point between performance and efficiency. The focus of this paper is more on how to make the DiT-based models achieve more stable training, higher inference quality and speed. In addition, we would like to highlight the additional contribution presented in our paper, as detailed in the Appendix. These include (1) zero-shot multilingual generation, (2) flexible visual recognition, and (3) text-to-multiview/music/audio/point clouds generation, all of which are yet to be explored for flow-based diffusion transformers.

---

### Official Review · Reviewer_ki5b · 2024-07-13

**Soundness:** 3
**Presentation:** 3
**Contribution:** 2
**Rating:** 4
**Confidence:** 4

**Summary:**

The paper introduces a new multi-modal generation model, Lumina-Next, which extends the previous Lumina-T2X approach by several innovations:  i) 3D rotary position embedding (RoPE) ii) extra normalization to stablize training iii) frequency- and time- aware scaled RoPE for training free resolution extrapolation iv) improved time scheduling and higher-order ODE solvers for reverse sampling. Lumina-Next aims to achieve faster, better, and more efficient generation on multiple text-to-X tasks, e.g. text-to-image/audio/point-cloud, etc.

**Strengths:**

Topic: the paper studies multi-modal generation using a unified framework, which would be of great interest to the community. The paper also promises to release the source code to support future research

Methodology: the proposed technical extensions for Lumina-T2X are technically sound.

Experiment: Lumina-Next demonstrates promising qualitative results on text-to-image generation, training-free resolution extrapolation, and text-to-point-cloud generation. The paper also provides quantitative evaluation on text-to-audio/music generation.

**Weaknesses:**

The reviewer appreciates the authors’ effort on improving text-to-X generation. However, as a technical submission for NeurIPS, the paper is fully convincing as it provides limited quantitative evaluation.

As a representative reader, the reviewer would expect

i) quantitative evaluations on all generation tasks

ii) extra apples-to-apples evaluations on any resolution recognition, as for now, there is only one baseline, DeiT-Base. The comparison is also apples-to-oranges as the models have different architectures and were trained on different data.

iii) comprehensive quantitative ablations on the proposed technical extensions, a) 3D RoPE,  b) extra normalization, c) frequency- and time- aware scaled RoPE, d) improved time scheduling and higher-order ODE solvers for reverse sampling, etc

**Questions:**

Please refer to the weaknesses

**Limitations:**

Limitations have been discussed in the Appendix

---

> ### Author Rebuttal · Authors · 2024-08-07
>
> ### Q1: Quantitative evaluations on all generation tasks.
>
> Thank you for the suggestion. We have further supplemented the quantitative comparison experiments. Due to time constraints, we only compared Lumina-Next with representative SoTA T2I models, SD3 and PixArt. To better illustrate the aesthetics and semantic ability, we conduct an AI preference study to evaluate Lumina-Next against other text-to-image models, since conventional metrics such as FID and CLIP-Score may not accurately reflect the generation quality. Following PixArt, we employ GPT-4o, the SoTA multimodal LLM exhibiting strong alignment with human preference, as our evaluator to vote based on image quality. As shown in the following table, Lumina-Next demonstrates competitive performance with advanced text-to-image models including PixArt and SD3. Note that SD3 uses over 1B text-image pairs, which is ~100x greater than our models, PixArt, which is already a training-efficiencient model, still uses 3x training compute than ours. However, we have to admit that Lumina-Next still underperforms these SoTA models in terms of text-image alignment and compositional generation, due to inadequate data and training.
>
> | Model | Winrate |
> | --- | --- |
> | SD3 | 69.5% |
> | PixArt | 43.6% |
>
> ### Q2: Extra apples-to-apples evaluations on any resolution recognition.
>
> Thanks for pointing this out! The lack of any resolution fine-tuning could indeed lead to unfair comparisons. To address this, we have added experiments with and without any resolution fine-tuning for any resolution inference. We also included two additional comparison methods: PVT-Base and Swin-Base. The experimental results show that under completely consistent settings, Next-DiT still achieves better performance over Deit-Base and PVT-Base, demonstrating the versatility of Next-DiT as a representation learning network for arbitrary resolutions. However, compared to Swin-Base, an advanced architecture with complicated architecture design for visual recognition tasks, Next-DiT still has room for improvement, objectively highlighting the superiority of specialized models in dedicated tasks. We have supplemented these results in the revised version of our paper.
>
> | Model | Size | Training Resolution | Inference Resolution | Results |
> | --- | --- | --- | --- | --- |
> | Deit-B | 86M | 224✕224 | Any-resolution  | 67.2 |
> | PVT-B | 61M | 224✕224 | Any-resolution  | 71.6 |
> | Swin-B | 88M | 224✕224 | Any-resolution  | 74.1 |
> | Next-DiT-B | 86M | 224✕224 | Any-resolution  | 72.6 |
> | Deit-B | 86M | 224✕224 + Any-resolution Fine-tuning | Any-resolution  | 82.9 |
> | PVT-B | 61M | 224✕224 + Any-resolution Fine-tuning | Any-resolution  | 83.2 |
> | Swin-B | 88M | 224✕224 + Any-resolution Fine-tuning | Any-resolution  | 85.3 |
> | Next-DiT-B | 86M | 224✕224 + Any-resolution Fine-tuning | Any-resolution  | 84.2 |
>
> ### Q3: Comprehensive quantitative ablations on the proposed technical extensions, a) 3D RoPE, b) extra normalization, c) frequency- and time- aware scaled RoPE, d) improved time scheduling and higher-order ODE solvers for reverse sampling, etc.
>
> Thanks for the advice! Due to time constraints, we are unable to provide all independent ablation experiments for 3D RoPE and additional normalization here. However, our results in Fig. 6 demonstrate that Next-DiT, the combination of these aforementioned techniques, achieved an overall improvement on the ImageNet benchmark compared to the Flag-DiT used by Lumina-T2X. Additionally, In the table below, we supplement quantitative ablation studies on the Lumina-Next T2I model to verify the effectiveness of our resolution extrapolation and inference acceleration techniques. We have added these results to our main paper.
>
> | Method | Resolution | CLIP Score | FID |
> | --- | --- | --- | --- |
> | Position Extrapolation | 2048✕2048 | 28.01 | 81.54 |
> | Position Interpolation | 2048✕2048 | 27.36 | 113.24 |
> | NTK-aware Scaled RoPE | 2048✕2048 | 28.21 | 61.89 |
> | Frequency-aware Scaled RoPE | 2048✕2048 | 28.52 | 59.92 |
> | Time-aware Scaled RoPE | 2048✕2048 | 28.93 | 58.66 |
>
> | Method | Resolution | Steps | CLIP Score | FID |
> | --- | --- | --- | --- | --- |
> | Uniform | 1024✕1024 | 10 | 26.87 | 78.23 |
> | Rational | 1024✕1024 | 10 | 28.57 | 64.12 |
> | Sigmoid | 1024✕1024 | 10 | 28.39 | 63.40 |

---

### Official Review · Reviewer_FoVd · 2024-07-15

**Soundness:** 2
**Presentation:** 2
**Contribution:** 2
**Rating:** 5
**Confidence:** 3

**Summary:**

This paper introduces the next generation of Lumina-T2X, Lumina-Next, which offers improved architecture, a scaled dataset, optimized sampling techniques, and a more efficient context extrapolation strategy. The improved architecture shows faster convergence rates, while the optimized sampling technique enables high-quality text-to-image generation with fewer steps. Through visual results, the authors validate the improvements provided by refining Flag-DiT in Lumina-T2X.

**Strengths:**

* Next-DiT brings excellent high-resolution extrapolation generation.

* Compared to previous work, it provides better generation quality under a few-step generation setting.

**Weaknesses:**

* There is a lack of quantitative experiment comparison results on some Text2Image benchmarks to demonstrate the superiority and effectiveness of the proposed method. Most of the comparisons are shown through visualization.

**Questions:**

* Text-to-image benchmark result should be evaluated and reported

---

> ### Author Rebuttal · Authors · 2024-08-07
>
> ### Q1: There is a lack of quantitative experiment comparison results on some Text2Image benchmarks to demonstrate the superiority and effectiveness of the proposed method.
>
> Thank you for the suggestion. We have further supplemented the quantitative comparison experiments.
>
> Due to time constraints, we only compared Lumina-Next with representative SoTA T2I models, SD3 and PixArt. To better illustrate the aesthetics and semantic ability, we conduct an AI preference study to evaluate Lumina-Next against other text-to-image models, since conventional metrics such as FID and CLIP-Score may not accurately reflect the generation quality. Following PixArt, we employ GPT-4o, the SoTA multimodal LLM exhibiting strong alignment with human preference, as our evaluator to vote based on image quality. As shown in the following table, Lumina-Next demonstrates competitive performance with advanced text-to-image models including PixArt and SD3. Note that SD3 uses over 1B text-image pairs, which is ~100x greater than our models, PixArt, which is already a training-efficiencient model, still uses 3x training compute than ours. However, we have to admit that Lumina-Next still underperforms these SoTA models in terms of text-image alignment and compositional generation, due to inadequate data and training.
>
> | Model | Winrate |
> | --- | --- |
> | SD3 | 69.5% |
> | PixArt | 43.6% |
>
> In addition to the ImageNet experiments in our paper, we also conduct various ablation studies to verify the effectiveness of each proposed component.
>
> | Method | Resolution | CLIP Score | FID |
> | --- | --- | --- | --- |
> | Position Extrapolation | 2048✕2048 | 28.01 | 81.54 |
> | Position Interpolation | 2048✕2048 | 27.36 | 113.24 |
> | NTK-aware Scaled RoPE | 2048✕2048 | 28.21 | 61.89 |
> | Frequency-aware Scaled RoPE | 2048✕2048 | 28.52 | 59.92 |
> | Time-aware Scaled RoPE | 2048✕2048 | 28.93 | 58.66 |
>
> | Method | Resolution | Steps | CLIP Score | FID |
> | --- | --- | --- | --- | --- |
> | Uniform | 1024✕1024 | 10 | 26.87 | 78.23 |
> | Rational | 1024✕1024 | 10 | 28.57 | 64.12 |
> | Sigmoid | 1024✕1024 | 10 | 28.39 | 63.40 |

---

### Official Review · Reviewer_NAeC · 2024-07-19

**Soundness:** 3
**Presentation:** 3
**Contribution:** 2
**Rating:** 5
**Confidence:** 4

**Summary:**

Lumina-T2X encounters challenges including training instability, slow inference, and extrapolation artifacts. This paper introduces Lumina-Next, which improves Lumina-T2X with improved architecture, scaled dataset, optimized sampling techniques, and better context extrapolation strategy.

On the architecture side, they replace 1D RoPE with 3D RoPE to eliminate the inappropriate positional priors to model images and videos using the attention mechanism. They further removed all learnable identifiers in Flag-DiT, and introduced the sandwich normalization block in attention modules to control the activation magnitudes.

On the context extrapolation side, the authors propose a Frequency-Aware Scaled RoPE, reducing content repetition during extrapolation, and a novel Time-Aware Scaled RoPE for diffusion transformers to generate high-resolution images with global consistency and local details.

On the sample techniques side, the authors propose time schedules tailored for flow models to minimize discretization errors. They further combine the optimized schedules with higher-order ODE solvers, achieving high-quality text-to-image generation samples within 5 to 10 steps.

With the improved architecture, context extrapolation, and fast sampling techniques, the authors show that Lumina-Next yields strong generation capabilities, such as generating high-quality images much larger than its training resolution and multilingual text-to-image generation.

Furthermore, they extend Lumina-Next’s versatility to other modalities, such as multiviews, audio, music, and point clouds, with minimal modifications, achieving superior results across these diverse applications.

**Strengths:**

- The paper is well written with extensive results, applications and details, which is valuable to the community.
- The paper provides practical improvements over architecture, dataset, sampling techniques, and context extrapolation strategy.
- The rethinking of existing model design is appreciated, especially showing a much smaller network can get better performance.

**Weaknesses:**

- the changes over Lumina-T2X are a bit incremental, considering Lumina-T2X is just released a month before the submission. Some of the designs in Lumina-T2X itself such as learnable tokens are not essential and only exist in Lumina-T2X, so the improvement (removing it) is not applicable to other DiTs, so removing extra learnable tokens itself can hardly be considered innovation.
- the 3D rope does not show advantage over native designs, such as normal 2D + temporal positional embedding. There is no application showing the usage of 3D RoPE, such as video generation.
- the ImageNet experiments can hardly justify the improvements. The comparison between Lumina-Next and SiT/Flag-DiT is not clear, such as whether the model sizes are the same.
- there is a lack of explanation why model size can be reduced from the Flag-DiT 5B/7B model significantly.
- The adoption of newer and stronger text encoders is also not ablated, so the improvement over Lumia-T2X cannot fully be attributed to the proposed changes.
- There is a lack of quadratic evaluation metrics such as TIFA and ImageReward scores for fair comparison across models.

**Questions:**

With all those changes on controlling the activation magnitudes, what about simply using grad clip norm to reduce gradient norms?

**Limitations:**

There are more interesting contents in the appendix, which can be moved to the main paper.

---

> ### Author Rebuttal · Authors · 2024-08-07
>
> ### Q1: Incremental Changes.
>
> Good point! We agree that Lumina-Next is largely based on Lumina-T2X with several improvements. However, we would like to highlight that these changes are made after comprehensive examinations and are essential for scaling this flow-based diffusion transformer or applying it in diverse fields. For example, we discover the exploding network activation and then propose to use sandwich normalization, which effectively eliminates the instability during both training and inference. In addition, we would like to highlight the additional contribution presented in our paper, as detailed in the Appendix. These include (1) zero-shot multilingual generation, (2) flexible visual recognition, and (3) text-to-multiview/music/audio/point clouds generation, all of which are yet to be explored for flow-based diffusion transformers.
>
> ### Q2: The advantage of 3D rope.
>
> Well said! The current implementation of 3D RoPE is almost equivalent to normal 2D RoPE + temporal positional embedding. We use 3D RoPE to continue one of the excellent features of Lumina-T2X — a unified framework for different modalities. For example, we adopt the 3D RoPE for multiview generation as a natural extension, since it can be treated as a sequence of image frames. The high-quality and consistent multiview results shown in Figure 25 verified the effectiveness of 3D RoPE in modeling 3D signals. Due to current limitations in training resources and data availability, Lumina-Next is still a unified multimodal framework with independently trained models. However, we consider a unified multimodal generative model as a necessary future step, so this unified framework can serve as an essential foundation for the next stage.
>
> ### Q3: The ImageNet experiments can hardly justify the improvements.
>
> Sorry for the confusion. Our comparison in Figure 5 is completely fair — with the same parameter settings and model sizes. Besides, the results from the ImageNet benchmark are also widely regarded as consistent when model scaling up and are commonly used as a preliminary validation test to reduce trial-and-error costs, e.g., EDM [3] conducts extensive experiments on ImageNet to gain valuable insights, which are further proven to be transferrable to various scales and fields.
>
> [3] Karras, Tero, et al. "Elucidating the design space of diffusion-based generative models." Advances in neural information processing systems 35 (2022): 26565-26577.
>
> ### Q4:  Lack of explanation about model size.
>
> Thanks for posing this question. Since the scale of data we use is much smaller than some open-source models (e.g., 30M in Lumina-Next V.S. over 1B in SD3), we found that, given the current data scale, the benefits of further scaling up the model to more parameters are not significant. This aligns with the finding in Lumina-T2X, where the 3B and 7B models achieved similar results. To achieve a sweet point between performance and efficiency, we adopted a 2B model, making it more accessible for contributors to run Lumina-Next on their personal devices. We have added more explanation in the revised version of our paper.
>
> ### Q5: The adoption of Gemma.
>
> Please allow us to clarify that we chose Gemma 2B mainly for better efficiency, and it is not considered as one of our contributions. Actually, Gemma 2B is weaker than LLaMA 7B used in Lumina-T2X. The influence of using different text encoders is valuable and worth exploring. However, this is not the major focus of our paper. Besides, the effectiveness of our core contributions has been well validated as follows: (i) the experiments on the ImageNet benchmark demonstrate the improvements of Next-DiT over Flag-DiT, and (2) fair experiments on the Lumina-Next T2I model have confirmed the effectiveness of our resolution extrapolation and inference acceleration techniques. We have further supplemented the quantitative results of ablation studies here.
>
> | Method | CLIP Score | FID |
> | --- | --- | --- |
> | Position Extrapolation | 28.01 | 81.54 |
> | Position Interpolation | 27.36 | 113.24 |
> | NTK-aware Scaled RoPE | 28.21 | 61.89 |
> | Frequency-aware Scaled RoPE | 28.52 | 59.92 |
> | Time-aware Scaled RoPE | 28.93 | 58.66 |
>
> | Method | Steps | CLIP Score | FID |
> | --- | --- | --- | --- |
> | Uniform | 10 | 26.87 | 78.23 |
> | Rational | 10 | 28.57 | 64.12 |
> | Sigmoid | 10 | 28.39 | 63.40 |
>
> ### Q6: Lack of quadratic evaluation metrics.
>
> Thank you for the suggestion. Due to time constraints, we only compared Lumina-Next with representative SoTA T2I models, SD3 and PixArt. To better illustrate the aesthetics and semantic ability, we conduct an AI preference study to evaluate Lumina-Next against other text-to-image models, since conventional metrics such as FID and CLIP-Score may not accurately reflect the generation quality. Following PixArt, we employ GPT-4o as our evaluator to vote based on image quality. As shown in the following table, Lumina-Next demonstrates competitive performance with advanced text-to-image models including PixArt and SD3. Note that SD3 uses over 1B text-image pairs, which is ~100x greater than our model, PixArt, which is already a training-efficiencient model, still uses 3x training compute than ours. However, we have to admit that Lumina-Next still underperforms these SoTA models in terms of text-image alignment and compositional generation, due to inadequate data and training.
>
> | Model | Winrate |
> | --- | --- |
> | SD3 | 69.5% |
> | PixArt | 43.6% |
>
> ### Q7: What about simply using grad clip norm.
>
> Good point! We have tried using techniques like gradient clip, but they are not that effective in preventing model training from crashing. This finding has also been mentioned in other papers such as ViT-22B [4]. We have included these empirical findings in the paper, hoping to provide some helpful insights to the community.
>
> [4] Dehghani, Mostafa, et al. "Scaling vision transformers to 22 billion parameters." *International Conference on Machine Learning*. PMLR, 2023.

---

### Author Response · Authors · 2024-08-14
**Request for Feedback on Rebuttal Points**

Dear Reviewers,

We would like to extend our appreciation for your time and valuable comments. We are eagerly looking forward to receiving your valuable feedback and comments on the points we addressed at the end of the rebuttal. Ensuring that the rebuttal aligns with your suggestions is of utmost importance.

Thank you for your dedication to the review process.

Sincerely,

Authors

---

### Decision · Program_Chairs · 2024-09-25

**Decision:**

Accept (poster)

**Comment:**

This is a “bag of tricks” style paper that proposes a set of techniques to improve upon the recent Lumina-T2X family of models. Although one can argue that these improvements are largely incremental on their own, there is still great value to testing and integrating them in a single system. Reviewers are generally positive about the contributions, and the AC believes that the authors did a good job in the rebuttal to address some of the weaknesses on the evaluation front. The AC leans towards accepting this work based on the promise that the code and model weights will be released, which should be of great value to the community.